# Diverse and complex muscle spindle afferent firing properties emerge from multiscale muscle mechanics

Kyle P Blum[1,2]*, Kenneth S Campbell[3], Brian C Horslen[2], Paul Nardelli[4], Stephen N Housley[4], Timothy C Cope[2,4], Lena H Ting[2,5]*

[1]Department of Physiology, Feinberg School of Medicine, Northwestern University, Chicago, United States; [2]Coulter Department of Biomedical Engineering, Emory University and Georgia Institute of Technology, Atlanta, United States; [3]Department of Physiology, University of Kentucky, Lexington, United States; [4]School of Biological Sciences, Georgia Institute of Technology, Atlanta, United States; [5]Department of Rehabilitation Medicine, Emory University, Atlanta, United States

**Abstract** Despite decades of research, we lack a mechanistic framework capable of predicting how movement-related signals are transformed into the diversity of muscle spindle afferent firing patterns observed experimentally, particularly in naturalistic behaviors. Here, a biophysical model demonstrates that well-known firing characteristics of mammalian muscle spindle Ia afferents – including movement history dependence, and nonlinear scaling with muscle stretch velocity – emerge from first principles of muscle contractile mechanics. Further, mechanical interactions of the muscle spindle with muscle-tendon dynamics reveal how motor commands to the muscle (alpha drive) versus muscle spindle (gamma drive) can cause highly variable and complex activity during active muscle contraction and muscle stretch that defy simple explanation. Depending on the neuromechanical conditions, the muscle spindle model output appears to 'encode' aspects of muscle force, yank, length, stiffness, velocity, and/or acceleration, providing an extendable, multiscale, biophysical framework for understanding and predicting proprioceptive sensory signals in health and disease.

**\*For correspondence:**
kylepblum@gmail.com (KPB);
lting@emory.edu (LHT)

**Competing interests:** The authors declare that no competing interests exist.

## Introduction

Amongst the many somatosensory receptors throughout the body that give rise to our kinesthetic and proprioceptive senses, muscle spindles have a unique muscle-within-muscle design such that their firing depends critically on both peripheral and central factors. Muscle spindle sensory signals are shaped not only by the movements and forces on the body, but also by motor commands to both the muscle (alpha drive) and to specialized muscle fibers in the muscle spindle mechanosensory region (gamma drive). But, the neuromechanical interactions that lead to muscle spindle firing patterns in movement are still poorly understood. Over many decades, experimental studies of muscle spindles across several vertebrate species – from snakes to mammals and including humans – provide us with rich literature of muscle spindle firing patterns during well-controlled conditions, revealing significant nonlinearities and history-dependence with respect to muscle length and velocity (*Haftel et al., 2004*; *Matthews, 1981*; *Nichols and Cope, 2004*; *Prochazka and Ellaway, 2012*; *Proske and Gandevia, 2012*). Moreover, manipulations to gamma motor drive reveal that muscles spindle firing patterns do not uniquely encode whole-muscle length and velocity even in tightly controlled experimental conditions (*Crowe and Matthews, 1964a*; *Crowe and Matthews, 1964b*; *Jansen and Matthews, 1962*). Independent activation of alpha and gamma motor neurons show that they can each profoundly alter muscle spindle sensory signals, rendering a direct prediction of

muscle spindle signals from recordings of movement highly unlikely. Indeed, the few available recordings of muscle spindles during natural motor behaviors collectively exhibit and complex relationship(s) to the biomechanics of movement that defies simple explanation (*Ellaway et al., 2015*; *Prochazka et al., 1976*; *Prochazka et al., 1977*; *Prochazka and Ellaway, 2012*; *Prochazka and Gorassini, 1998b*; *Taylor et al., 2000*; *Taylor and Cody, 1974*; *Taylor et al., 2006*).

Developing a mechanistic framework for understanding how muscle spindle organs generate the complex sensory signals during natural movements is critical for understanding muscle spindle sensory signals and how they change under different behaviors, in neurological disorders, and during interactions with devices or interventions to improve motor impairments. Existing computational models of muscle spindle afferent firing have largely been data-driven, with well-known experimental features represented phenomenologically (*Schaafsma et al., 1991*; *Hasan, 1983*; *Lin and Crago, 2002*; *Mileusnic et al., 2006*). Such models have high fidelity under the conditions in which the data were collected, but generalize poorly to more naturalistic conditions. The anatomical arrangement of the muscle and muscle spindle have also been shown in simulation to affect the forces on the muscle spindle and alter muscle spindle firing properties (*Lin and Crago, 2002*). The biomechanical properties of the intrafusal muscle fibers within the muscle spindle have also been implicated in determining muscle spindle receptor potentials (*Blum et al., 2017*; *Fukami, 1978*; *Hunt and Ottoson, 1975*; *Hunt and Wilkinson, 1980*; *Nichols and Cope, 2004*; *Poppele and Quick, 1985*; *Proske and Stuart, 1985*), but intrafusal fiber forces have not been recorded experimentally in conjunction with muscle spindle afferent recordings, and only a few recordings of muscle spindle receptor potentials exist (*Fukami, 1978*; *Hunt et al., 1978*; *Hunt and Ottoson, 1975*; *Hunt and Wilkinson, 1980*).

Here, we take a biophysical modeling approach to build a muscle spindle model based on first principles of intrafusal muscle contractile mechanics, along with its interaction with the muscle and tendon to predict both classic and seemingly paradoxical muscle spindle Ia afferent firing during passive and active conditions. We first establish a relationship between muscle spindle firing characteristics and estimated intrafusal fiber force and its first time-derivative, yank (*Lin et al., 2019*), in passive stretch conditions. We then build a generative model based on a simple representation of intrafusal muscle mechanics, revealing that several classic muscle spindle firing characteristics emerge from muscle cross-bridge population kinetics. The independent effects of alpha and gamma drive on muscle spindle firing were then simulated based on mechanical interactions between the muscle spindle and the muscle-tendon unit. Highly variable muscle spindle activity observed during human voluntary isometric force generation and muscle stretch–from silent to highly active–could be explained by as a result of interactions between the muscle spindle and musculotendon unit, central activation of extrafusal (alpha drive) versus intrafusal muscle fibers (gamma drive), and muscle length change. These multiscale interactions may explain why muscle spindle activity may roughly approximate muscle force, length, velocity, acceleration, activation level, or other variables under different movement and experimental conditions. As such, our model establishes a biophysical framework to predict muscle spindle afferent activity during natural movements that can be extended to examine multiscale interactions at the level of cell, tissue, and limb mechanics.

## Results

### Muscle fiber force and yank reproduce diverse passive features of rat muscle spindle IFRs

We first demonstrated that in passive stretch of relaxed rat muscle, muscle spindle Ia afferent firing rates can be described in terms of extrafusal muscle fiber force and yank. We recorded muscle spindle afferent axonal potentials and computed instantaneous firing rates (IFRs) in anesthetized rats while stretching the triceps surae muscles (*Figure 1*). In the relaxed condition, that is in the absence of central drive to the muscles, we assumed that extrafusal muscle fiber forces provide a reasonable proxy for resistive forces of the intrafusal muscle fibers within the muscle spindle mechanosensory apparatus (*Figure 1*). We previously demonstrated that whole musculotendon force in rat experiments is not predictive of muscle spindle IFRs (*Blum et al., 2019*), and that only a portion of the total musculotendon force can be attributed to extrafusal muscle fiber force, with the remaining nonlinear elastic component attributed to extracellular tissues (*Blum et al., 2019*; *Meyer and Lieber,*

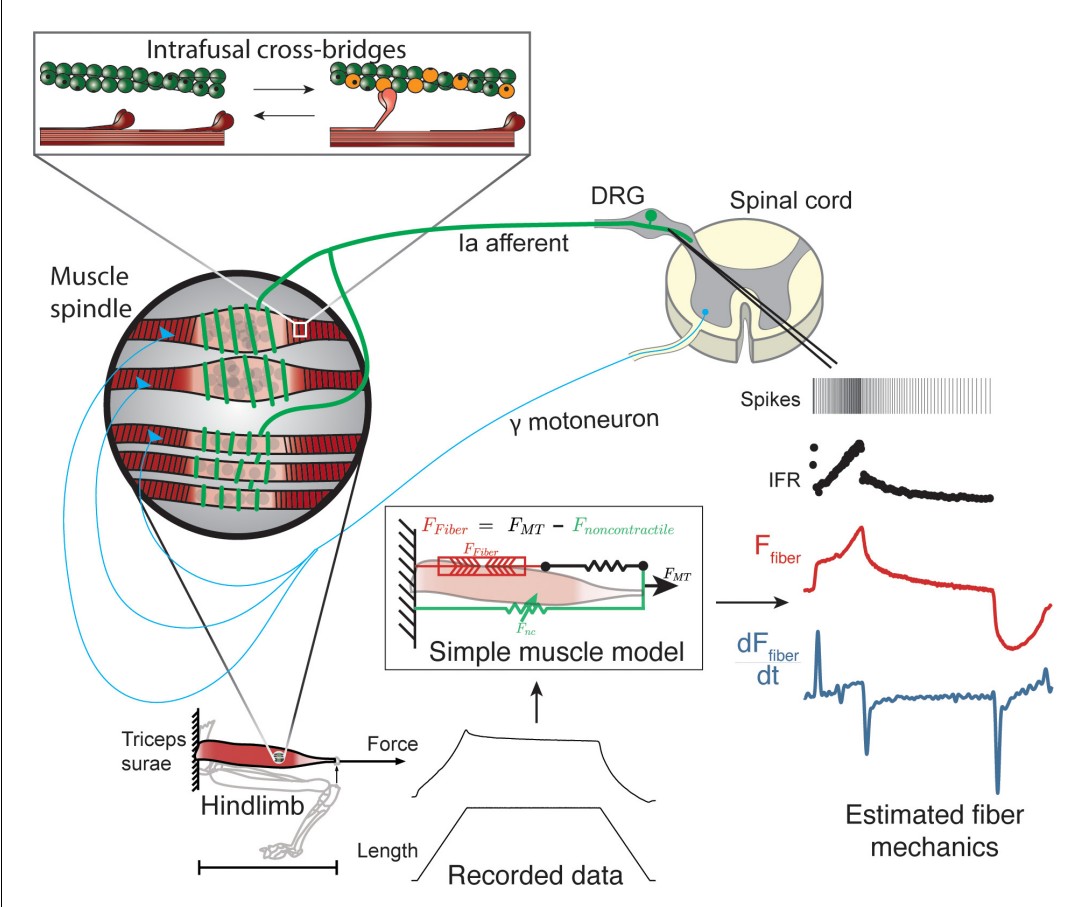

**Figure 1.** Overview of methodologies to test hypothesis that muscle spindle Ia afferent firing rates follow intrafusal muscle fiber force due to cross-bridge interactions. Ia afferent firing rates were recorded from dorsal rootlets during stretches of the triceps surae muscle in anesthetized rats. Muscle fiber forces were estimated by subtracting noncontractile forces from measured whole musculotendon force. The exponential rise in force with stretch was assumed to arise from non-contractile tissue in parallel with the muscle-tendon unit with exponential stiffness (*Blum et al., 2019*). The remaining estimated muscle fiber force and yank exhibited similar temporal characteristics to the muscle spindle IFR. Intrafusal muscle fiber force and yank were then simulated using a cross-bridge based model to predict muscle spindle IFRs.

*2011*; *Meyer and Lieber, 2018*). We modeled the extracellular tissue forces using a nonlinear elastic model and identified the parameters that minimized the prediction error of muscle spindle firing rates based on estimated extrafusal muscle fiber force and yank (see Materials and methods). The initial rise in extrafusal muscle fiber force at the onset of stretch (*Figure 1*, red trace) manifests as a large, transient yank signal (*Figure 1*, blue trace) and became more apparent once the extracellular tissue forces were subtracted from whole musculotendon force.

Like in cats (*Blum et al., 2017*), fine temporal details of rat Ia afferent firing in relaxed muscles could be reconstructed by linear combinations of estimated extrafusal muscle fiber force and yank, subjected to a threshold. We assumed average extrafusal fiber force to be proportional to intrafusal muscle fiber force in the anesthetized, passive stretch condition only. The notable features of muscle spindle Ia afferent firing reconstructions included initial bursts, dynamic responses during ramps, rate adaptation during holds, and movement history-dependent firing (*Figure 2A–C*; *Figure 2—figure supplement 1*). The reconstructions revealed history-dependent initial bursts in Ia afferent firing coincide with large, history-dependent yank component in estimated extrafusal muscle fiber force (*Figure 2B*). The dynamic Ia afferent firing response during ramp stretches was primarily reconstructed by the force component (*Figure 2C*) and was larger during the first stretch of a repeated sequence (*Figure 2B*). Rate adaptation during the hold period was reconstructed by the force component in both slow and fast stretches (*Figure 2C*; *Matthews, 1963*).

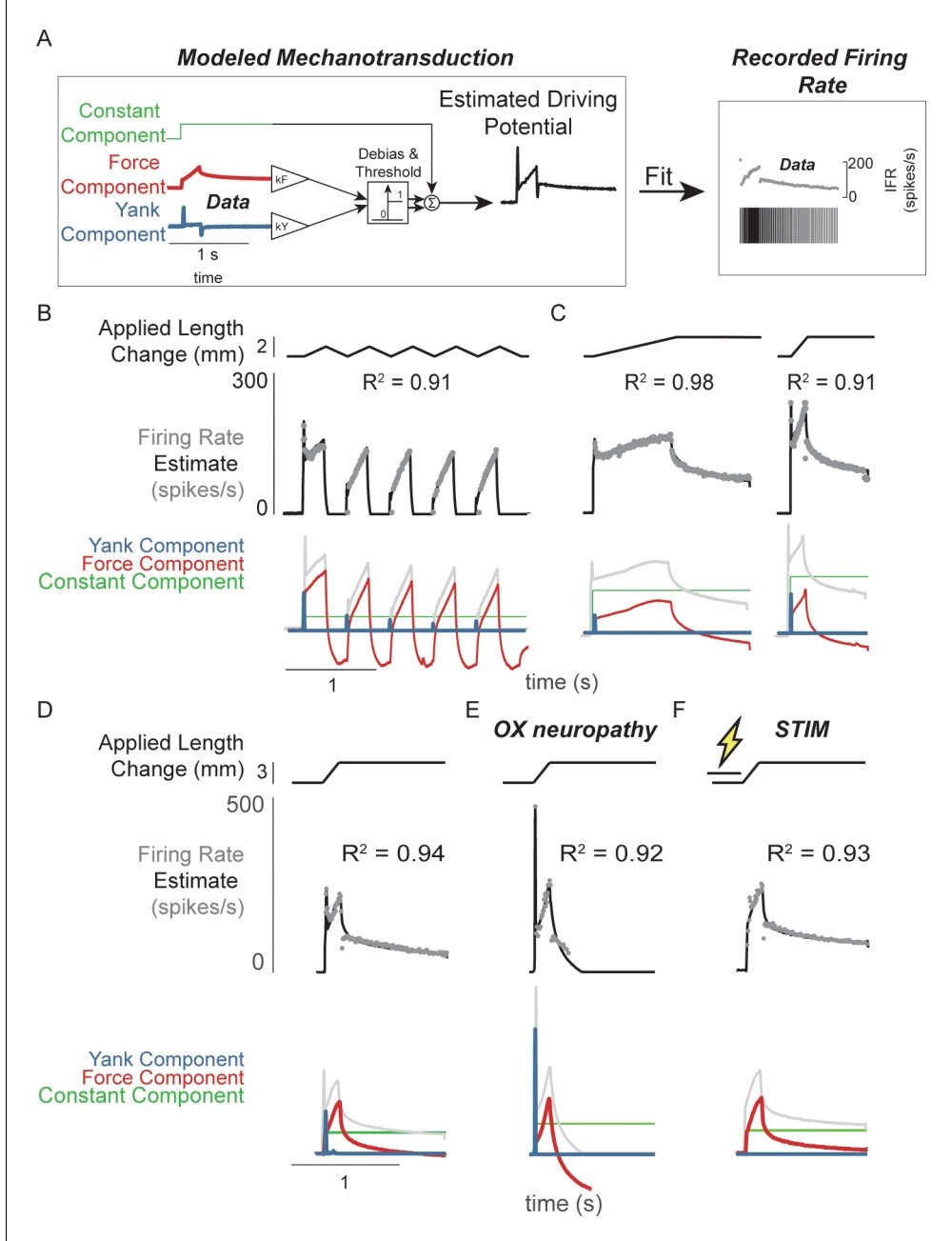

**Figure 2.** Muscle spindle driving potentials estimated from independent contributions of experimentally-derived muscle fiber force and yank. (**A**) Estimated driving potentials were derived from linear combinations of muscle fiber force and yank, half-wave rectified, and compared against recorded muscle spindle Ia afferent firing rates. Weights of each component were optimized to match recorded spiking dynamics. (**B**) Recorded muscle spindle Ia afferent firing rates (gray dots) in history-dependent conditions, having non-unique relationships to muscle length and velocity, were reproduced using muscle fiber force and yank (black lines). Notably, the initial burst and increased firing during ramp in the first stretch were attributed to increased muscle fiber yank and to greater force during the first stretch, respectively. (**C**) Likewise, muscle fiber force and yank could also account for the temporal dynamics of Ia afferent firing in response to both slow and fast stretches. (**D**) This model permits independent manipulation of the force and yank contributions to muscle spindle firing rates. As such, we can explain the altered muscle spindle Ia afferent firing patterns in (**E**) oxaliplatin chemotherapy-induced sensory neuropathy as a loss of force sensitivity, and (**F**) after antidromic electrical stimulation of the axon as loss of yank sensitivity.

The online version of this article includes the following figure supplement(s) for figure 2:

*Figure 2 continued on next page*

*Figure 2 continued*

**Figure supplement 1.** Goodness-of-fit to measured muscle spindle Ia afferent firing rates using estimated muscle fiber force and yank (blue) compared to kinematics model (red) as baseline comparison.

**Figure supplement 2.** Fits of muscle spindle firing rates before and after oxaliplatin-induced neuropathy using estimated muscle fiber force and yank.

**Figure supplement 3.** Estimated muscle fiber force-related model predicts changes in muscle spindle encoding caused by axonal stimulation.

We then demonstrated that the sensitivity of Ia firing to passive muscle fiber force and yank was differentially affected by two types of perturbation to the muscle spindle afferent neuron. Estimated extrafusal muscle fiber force- and yank-based reconstruction of Ia firing rates was robust to experimental perturbations due to either oxaliplatin chemotherapy (OX) alone or intra-axonal antidromic stimulation of the afferent (STIM). While the mechanisms underlying these perturbations are undetermined, the effects on firing likely involve alterations in function of ion channels in the nerve terminal function as opposed to effects on properties of non-neural tissues, for example muscle. (*Bullinger et al., 2011*; *Housley et al., 2020a*; *Housley et al., 2020b*). Accordingly, the characteristics of the estimated intrafusal muscle fiber force and yank were qualitatively similar in intact and OX rats, suggesting there was no change in muscle fiber force in the OX animals. However, muscle spindles in healthy rats treated with OX maintain an initial burst and dynamic response, but lack sustained firing during the hold period (*Figure 2D* vs *Figure 2E*; *Figure 2—figure supplement 2*; *Bullinger et al., 2011*). These OX Ia afferent firing phenotypes were primarily reconstructed by the yank component (*Figure 2D*, blue trace), with a small contribution of the force component (*Figure 2D*, red trace; *Figure 2—figure supplement 2*), suggesting a reduced sensitivity of the muscle spindle afferent to intrafusal muscle fiber force. Conversely, when we perturbed the Ia afferent in healthy rats through intra-axonal electrical stimulation (STIM; 500 ms duration, 100 Hz train of 30nA pulses) and applied muscle stretch immediately afterward (*Figure 2F*; *Figure 2—figure supplement 3*), STIM Ia afferent firing phenotypes were primarily reproduced by the intrafusal fiber force, with reduced sensitivity to the intrafusal fiber yank (*Figure 2—figure supplement 3*). Overall, we were able to reproduce perturbed Ia afferent firing data by varying only the relative weighting of force and yank. Taken together, these data show that in the absence of changes in intrafusal or extrafusal muscle fiber force and yank signals, the sensitivity of the muscle spindle Ia afferent to force and yank can be decoupled and therefore may arise due to separate encoding or transduction mechanisms.

## Modifying muscle spindle Ia afferent sensitivity to force and yank generates an array of firing phenotypes

The differential effects of oxaliplatin and axonal stimulation on spindle firing rates led us to hypothesize there is a degree of independence in the transduction of force and yank in the spindle. We further tested this hypothesis by simulating Ia afferent firing arising from force- and yank-based receptor driving potentials (*Figure 3A*; see Materials and methods). Nominal sensitivities of the model receptor current (closely related to driving potential) were chosen to reproduce a typical recorded muscle spindle firing rate during passive stretch (*Figure 3B*, green shaded box). We then generated a family of muscle spindle firing phenotypes (*Figure 3B*, blue dots) by systematically varying the sensitivity of receptor currents (*Figure 3*, thin black lines) to the same muscle fiber force (*Figure 3B*, vertical axis) and yank signals (*Figure 3B*, horizontal axis) during the same muscle stretch (*Figure 3B*).

Varying force- and yank- sensitivity generated diverse muscle spindle firing phenotypes similar to those observed experimentally – including the OX and STIM phenotypes. Under passive conditions, muscle spindle Ia afferent firing profiles exhibited larger initial bursts and dynamic responses as yank sensitivity increased (*Figure 3B*, left to right), consistent with classically identified dynamic muscle spindle firing phenotypes (*Emonet-Dénand et al., 1977*). High force sensitivity led to profiles with elevated plateau firing (*Figure 3B*, top to bottom), consistent with classically-identified static muscle spindle firing phenotypes (*Emonet-Dénand et al., 1977*; *Jansen and Matthews, 1962*, *Figure 3—figure supplement 1*). The firing profiles with high yank and lowest force sensitivity resembled the OX firing phenotype (*Bullinger et al., 2011*; *Figure 3B*, yellow shaded boxes, *Figure 2D*) and the firing profiles with lowest yank sensitivity resembled the STIM firing phenotype (*Figure 3B*, red

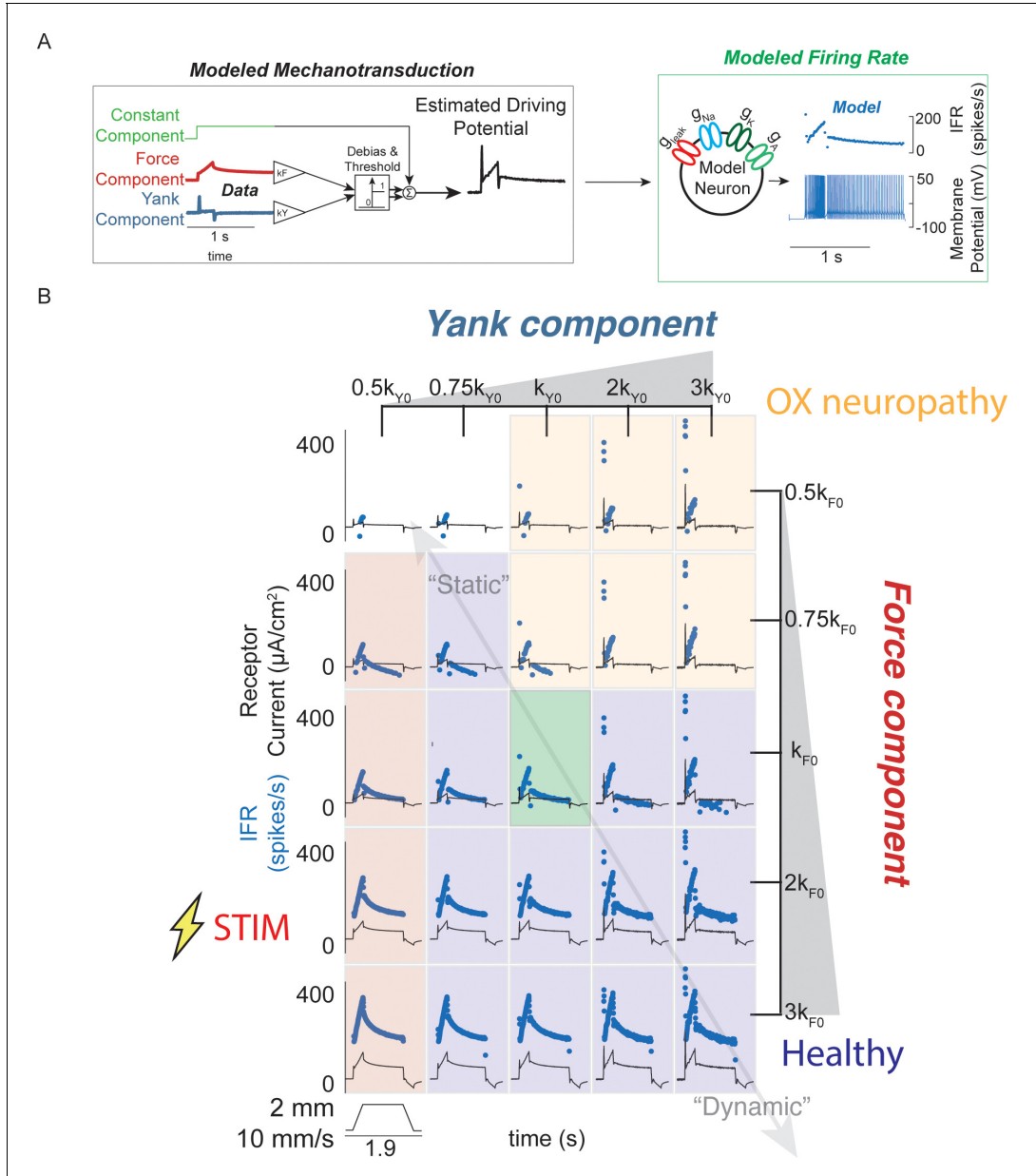

**Figure 3.** Spectrum of passive muscle spindle firing phenotypes accounted for by varying sensitivity to experimentally derived muscle fiber force and yank. (A) Sensitivity to experimentally-derived force and yank was systematically varied for a single stretch and resultant driving potentials were input to a Connor-Stevens model neuron to generate firing patterns. (B) Nominal force and yank weights were identified to recreate experimentally-recorded muscle spindle response to a representative stretch (green box). Increasing sensitivity to yank (left to right) generated larger initial bursts and dynamic responses during the ramp, and resembled responses from oxaliplatin-treated specimens at the highest yank and lowest force sensitivities (orange boxes, compare to *Figure 2E*). Increasing sensitivity to force (top to bottom) generated higher firing rates during the hold period and resembled Ia afferent firing responses after axonal stimulation at the lowest yank and highest force sensitivities (red boxes, compare to *Figure 2F*). Varying the weights of the force and yank sensitivities could recreate the spectrum of healthy muscle spindle firing profiles reported classically (purple boxes). The online version of this article includes the following figure supplement(s) for figure 3:

**Figure supplement 1.** Estimated muscle fiber force model predicts inter-afferent variability of healthy afferent firing properties across perturbation velocity and acceleration.

shaded boxes, *Figure 2D*). We concluded that independently varying Ia afferent sensitivity (the input-output gain) to force and yank explains a spectrum of spindle firing phenotypes.

## A muscle cross-bridge model predicts intrafusal muscle fiber force and yank underlying muscle spindle firing rates

We next built a biophysical model of muscle spindle mechanics to more directly predict intrafusal fiber force and yank during muscle stretch conditions where experimental data are not readily available. History-dependent muscle forces have been simulated previously based on muscle cross-bridge population cycling kinetics (*Campbell and Lakie, 1998*; *Campbell and Moss, 2000*; *Campbell, 2014*; *Campbell and Moss, 2002*), but no current muscle spindle model has incorporated these principles, and thus none simulate history dependence in muscles spindle Ia afferent firing (*Hasan, 1983*; *Lin and Crago, 2002*; *Mileusnic et al., 2006*; *Schaafsma et al., 1991*). Muscle spindles contain bundles of specialized 'bag' and 'chain' intrafusal muscle fibers (*Boyd, 1962*; *Boyd, 1976*; *Boyd et al., 1977*; *Gladden and Boyd, 1985*; *Taylor et al., 1999*). While different intrafusal muscle fibers vary morphologically and mechanically, as well as in their contributions to Ia firing patterns, their basic architectures are similar (*Banks et al., 1997*; *Banks, 2005*; *Banks, 2015*; *Gladden and Boyd, 1985*). Intrafusal muscle fibers are innervated by gamma motor neurons in two polar regions containing contractile muscle fibers, with an in-series non-contractile equatorial region around which the muscle spindle endings are wrapped and mechanotransduction occurs. Muscle spindle Ia receptor potentials and afferent firing are directly related to deformation of the equatorial regions (*Boyd, 1976*; *Boyd et al., 1977*), which depends on the tensile forces in intrafusal muscle fibers.

Our mechanistic muscle spindle model consists of a pair of half-sarcomere muscle models arranged in parallel, simulated using two-state actin-myosin population interactions (*Campbell, 2014*). The simulated 'dynamic fiber' loosely represented the putative dynamic bag1 intrafusal muscle fibers, thought to mediate muscle spindle initial bursts (*Banks et al., 1997*; *Proske and Stuart, 1985*; *Schäfer and Kijewski, 1974*; *Schäfer and Schäfer, 1969*; *Song et al., 2015*). The simulated 'static fiber' loosely represented the bag2 and chain intrafusal muscle fibers of the muscle spindle thought to mediate tonic muscle spindle firing (*Figure 4A*; *Poppele et al., 1979*). The dynamic fiber model was designed with slower myosin attachment rates (*Thornell et al., 2015*) and with a more compliant passive element than the static fiber (*Gladden, 1976*) (see Materials and methods; *Figure 4—figure supplement 1*). Length changes to the spindle were applied to both fibers equally. We assumed the receptor driving potential of the Ia afferent receptor ending to be proportional to a weighted combination of force and yank of the intrafusal fibers based on visual inspection of intrafusal force recordings and our previous hypotheses of intrafusal force and yank encoding (*Blum et al., 2019*; *Fukami, 1978*; *Hunt and Ottoson, 1975*). This was based on the idea that the intrafusal mechanosensory nuclear regions stretch linearly with intrafusal force (*Matthews, 1981*; *Schaafsma et al., 1991*) and do not themselves contribute significantly to the force profile of the intrafusal muscle fiber (*Figure 4A*, lower panel; see Materials and methods for more details).

Our mechanistic model predicted a spectrum of muscle spindle firing phenotypes identified in passive stretch conditions by varying the sensitivity (gain) of driving potentials to biophysically predicted intrafusal fiber force and yank (*Figure 4B*). The predicted muscle spindle firing phenotypes closely resembled biological firing phenotypes discussed earlier. As a result of our chosen kinetic scheme, history-dependent forces emerge within our simulated intrafusal muscle fibers from cross-bridge population dynamics (*Figure 4C*). When the muscle is at rest, the spontaneous formation and breakage of cross-bridges is at steady-state equilibrium (*Figure 4C*, time 1). As soon as a stretch is applied, the population of attached cross-bridges is also stretched, resulting in a short period of high-stiffness force response, known as short-range stiffness (*Figure 4C*, time 2; *Lakie and Campbell, 2019*). Once these cross-bridges are stretched to their limits, they break and a new attachment equilibrium is reached during the remainder of stretch (*Figure 4C*, between times 2 and 3). If the muscle is shortened back to the original length, the cross-bridges will shorten (*Figure 4C*, time 4 – note the distributions shift leftward). In this case, the force in the fibers will decrease until the fibers are slack (*Figure 4C*, time 5), at which point the cross-bridges will shorten the muscle fibers against zero load until the slack length is reduced to zero, or another stretch is applied (*Figure 4C*, time 6). If the fibers are not given enough time to return to steady-state

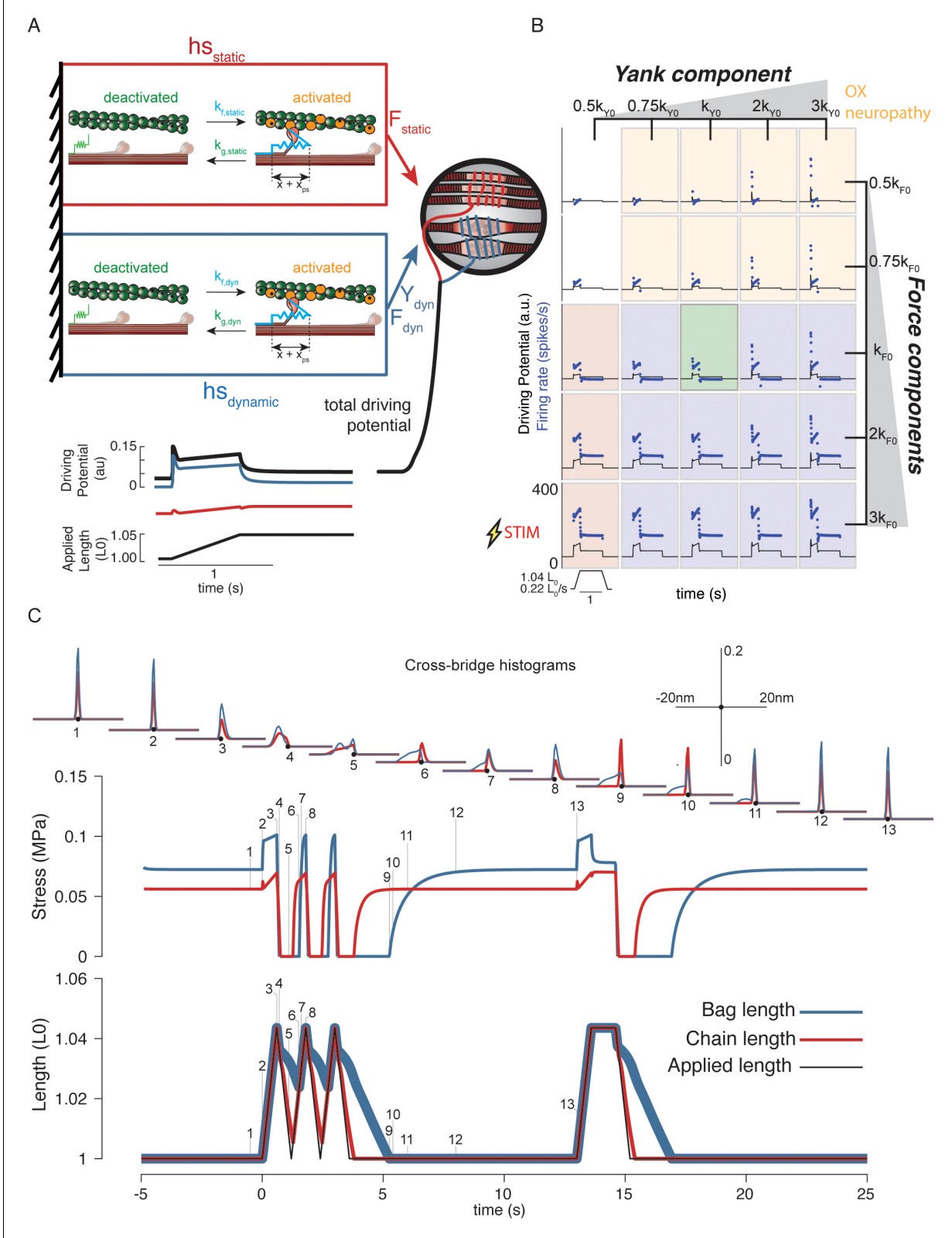

**Figure 4.** Generative model of muscle spindle driving potentials based on simulated muscle cross-bridge kinetics. (**A**) The muscle spindle model consists of two muscle fibers in a parallel mechanical arrangement, loosely representing intrafusal bag and chain fibers. (**B**) During stretch, force and yank of the 'dynamic' fiber is linearly combined with force of the 'static' fiber, with different proportions generating driving potentials consistent with 'dynamic' and 'static' muscle spindle firing response phenotypes. (**C**) A population of myosin cross-bridges and their relative displacement and velocity

*Figure 4 continued on next page*

Figure 4 continued

with respect to active actin binding sites was simulated during three consecutive ramp-stretch-release stretches. The distribution of cross-bridge lengths relative to actin binding sites is shown at different timepoints of imposed kinematics (numbered graphs and timepoints). The length of the dynamic and static fibers (lower trace) and the stress in the dynamic and static fibers (middle trace) is shown. Deviations between applied length and muscle fiber length occur due to muscle fiber slack.

The online version of this article includes the following figure supplement(s) for figure 4:

**Figure supplement 1.** Biophysical intrafusal muscle models.

equilibrium (e.g. *Figure 4C*, time 1), a stretch will result in qualitatively different, history-dependent force and yank responses.

## A biophysical muscle spindle model predicts a variety of classic muscle spindle firing phenomena

A truly general model of a muscle spindle would predict firing properties across different experimental contexts without any fundamental changes to the model structure or parameters. Using a single, nominal set of parameters in our muscle spindle model, we tested whether a variety of classical muscle spindle firing characteristics during passive stretch would emerge, including: fractional power relationship with stretch velocity (*Matthews, 1963*), history-dependence (*Haftel et al., 2004*), and increased firing due to gamma motor neuron activation (*Boyd et al., 1977*; *Crowe and Matthews, 1964a*; *Crowe and Matthews, 1964b*). To demonstrate the ability of the mechanical signals themselves to predict the properties observed in Ia afferent firing rates, we examined simulated driving potentials rather than creating additional degrees of freedom in our model with a spike generating process.

In our model, the scaling properties of muscle spindle dynamic responses and initial bursts with increasing velocity during passive muscle stretch arise from intrafusal cross-bridge kinetics. More specifically, the strain dependence of the cross-bridge detachment rates produces force profiles that contain linear increases in short-range stiffness and sublinear increases in force dynamic response as a function of stretch velocity. In a series of constant velocity stretches relative to optimal muscle length, $L_0(0.1–1.0\ L_0/s$ in 0.1 $L_0/s$ increments; *Figure 5A*), we computed the 'dynamic index', classically defined as the difference in firing rate between the end of the ramp and after 0.5 s of isometric hold, which increases with stretch velocity (*Figure 5B–C*; *Matthews, 1963*). We predicted a sublinear increase in dynamic index with stretch velocity, emergent from intrafusal mechanics, resembling the classically reported fractional-power velocity relationship in muscle spindle firing rates (*Figure 5C*; *Houk et al., 1981*). In the same simulations, linear scaling of initial burst amplitude with stretch acceleration was predicted (*Schäfer and Kijewski, 1974*; *Schäfer and Schäfer, 1969*). Initial burst scaling was emergent from intrafusal muscle fiber yank at the onset of stretch due to the elasticity of attached cross-bridges that then detach rapidly after being stretched a small fraction of $L_0$ (*Hasan and Houk, 1975*; *Matthews and Stein, 1969*). To our knowledge, neither of these phenomena has been previously demonstrated to arise from the same mechanistic model.

Our biophysical model predicted history-dependent changes in the muscle spindle firing initial burst and dynamic response analogous to those previously reported in acute cat, rat, and toad experiments as well as in microneurographic recordings in awake humans (*Blum et al., 2017*; *Day et al., 2017*; *Haftel et al., 2004*; *Proske and Stuart, 1985*). In our model, these features emerged from the asymmetry present in strain-dependent cross-bridge detachment rates. In three consecutive, identical stretches, the biophysical muscle spindle model predicted an initial burst and elevated dynamic response on the first, but not second stretch (*Figure 6A*). In the third stretch, the amplitude of the simulated driving potentials recovered asymptotically as the time interval between stretches increased to 10 s (*Figure 6A*), resembling the recovery of spike counts during the dynamic response in rats (*Figure 6B*; *Haftel et al., 2004*) and the recovery of initial bursts as a function of hold period in toads (*Figure 6B*; *Proske and Stuart, 1985*). Our model predicted an initial burst at the onset of sinusoidal muscle stretches (*Figure 6C*), similar to that seen in microneurographic recordings from awake humans (*Figure 6C*; *Day et al., 2017*). Although a cross-bridge mechanism has been proposed previously based on experimental findings (*Haftel et al., 2004*; *Nichols and*

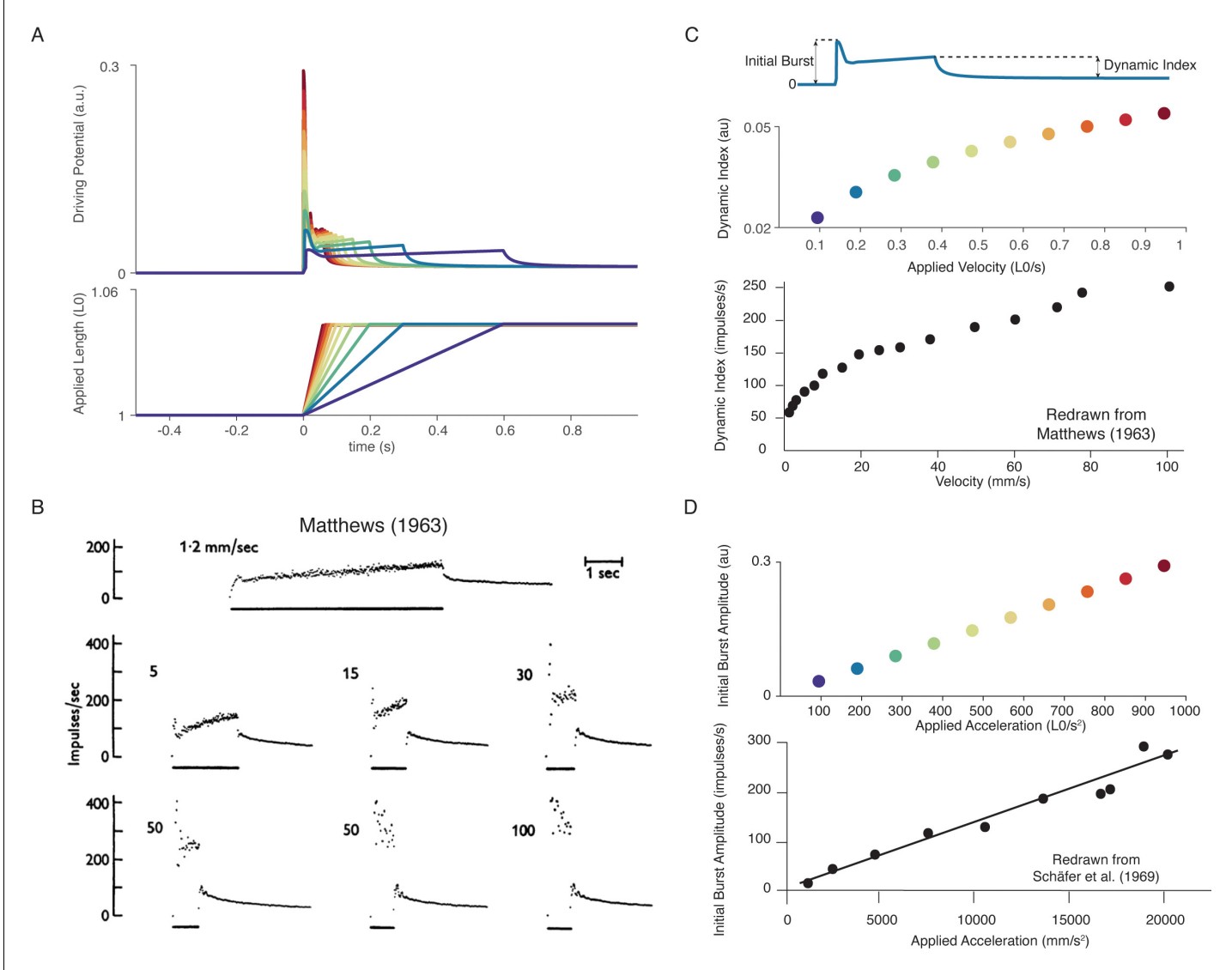

**Figure 5.** Muscle spindle firing dynamics emerge from cross-bridge mechanisms during simulated muscle stretch. Simulations assume a low level of muscle cross-bridge cycling consistent with relaxed muscle. Length displacements were imposed on the muscle fiber. (A) Predicted driving potentials (upper traces) during ramp stretches of varying velocity and acceleration (lower traces). (B) Classical data showing Ia afferent firing modulation with different stretch velocity (***Matthews, 1963***). (C) Dynamic index emergent from cross-bridge mechanisms. Dynamic index is defined classically as the ratio of firing rate at the end of the ramp phase and the firing rate 0.5 s into the hold phase (upper diagram). The muscle spindle model exhibits a sublinear relationship between dynamic index and stretch velocity (middle plot – colors correspond to A), similarly to classical findings (bottom plot). (E) Linear acceleration scaling of initial burst emergent from cross-bridge mechanisms. Initial burst amplitude is defined as the difference between peak firing rate during initial burst and baseline. Muscle spindle model exhibits linear scaling with stretch acceleration at stretch onset (top plot), which is consistent with classical findings (bottom plot – ***Schäfer and Schäfer, 1969***). ***Figure 5B,C*** (bottom panel) is reproduced from ***Matthews and Stein, 1969*** with permission from John Wiley and Sons. These images are not covered by the CC-BY 4.0 license and further reproduction of these panels would need permission from the copyright holder. ***Figure 5D*** (bottom panel) is redrawn from ***Schäfer and Schäfer, 1969*** with permission from Springer Nature.

***Cope, 2004***; ***Proske et al., 1992***; ***Proske and Gregory, 1977***; ***Proske and Stuart, 1985***), this is the first demonstration of history dependence in a muscle spindle model.

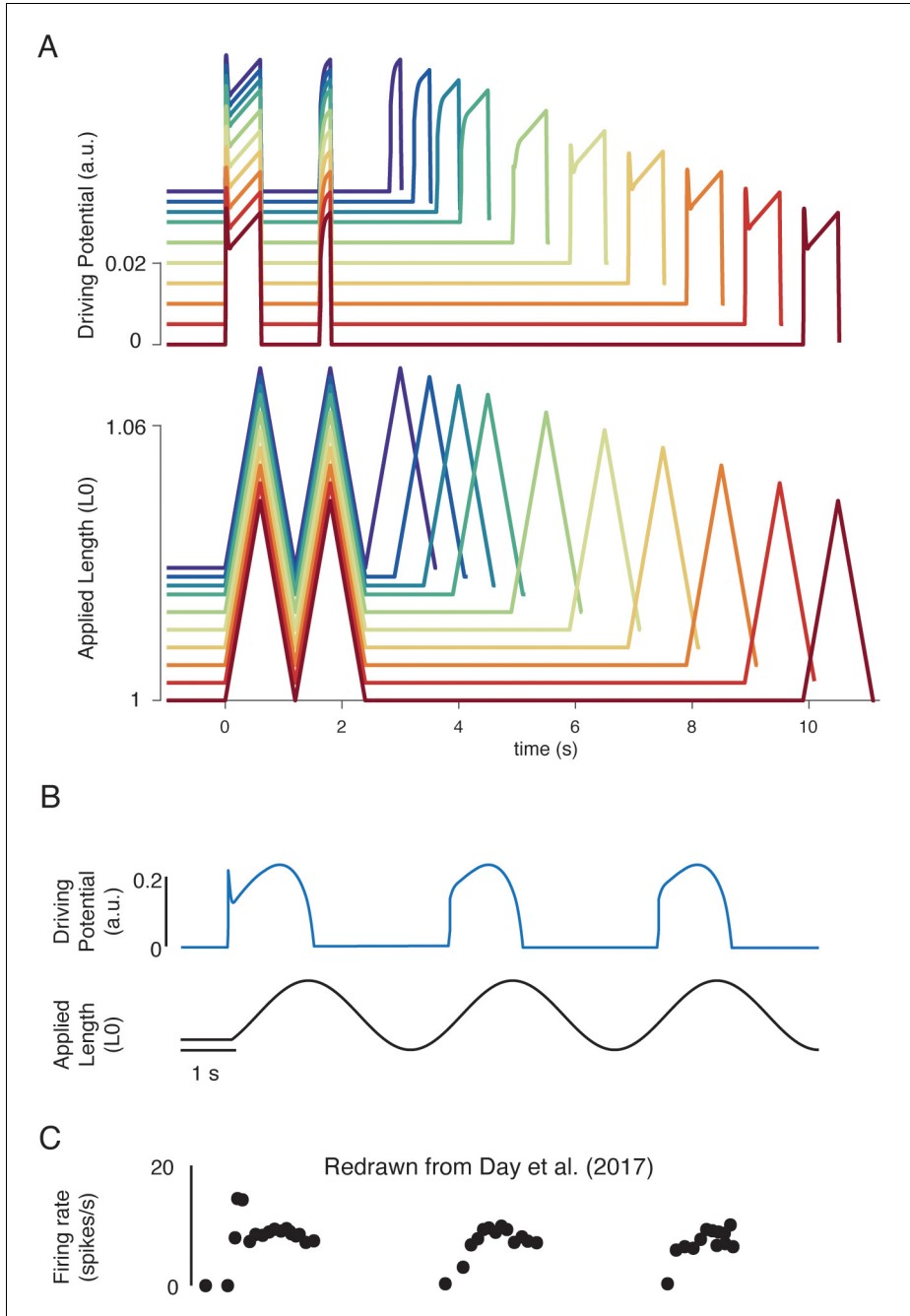

**Figure 6.** Emergent movement history-dependence of muscle spindle. (**A**) When repeated stretch-shorten length cycles were applied, a larger response was predicted if the length was held constant prior to stretch (bottom plot – all traces). An abolished initial burst and reduced dynamic response were predicted in the second stretch, immediately applied after the length was returned to the initial value (top plot – all traces). In the third stretch, recovery of the initial burst was dependent on the time interval between the second and third stretch, with the effect saturating between 5 and 10 s (top plot, recovery from violet to red traces). This finding predicts the results found by *Haftel et al., 2004* in rat muscle spindles. Similarly, dynamic response recovered gradually with time interval between second and third stretch (top plot, recovery from violet to red traces). This finding predicts the results found by *Proske and Stuart, 1985* in toad muscle spindles. (**B**) Sinusoidal displacement imposed from rest elicited a history-dependent initial burst in the predicted muscle spindle driving potential at the onset of stretch, resembling data from (**C**) human muscle spindles recorded during the application of sinusoidal motion to the ankle in relaxed conditions (*Day et al., 2017*). *Figure 6C* is redrawn from *Day et al., 2017* with permission from The American Physiological Society.

## A biophysical muscle spindle model predicts effects of gamma drive on muscle spindle firing

We tested whether previously-reported effects of gamma motor neuron activity on muscle spindle afferent firing characteristics during passive muscle stretch conditions were also emergent from our biophysical model. Dynamic and static gamma motor neurons innervate the static and dynamic intrafusal muscle fibers within the muscle spindle, respectively. To simulate classic experiments in which dynamic and static gamma motor neurons were electrically stimulated in anesthetized animals, we independently increased the number of available actin-binding sites in simulated intrafusal static and dynamic muscle fibers (*Figure 7A*). Consistent with many prior experimental findings, simulated dynamic gamma drive increased the force and yank of the dynamic fiber during stretch, predicting increased receptor driving potentials underlying initial burst and dynamic responses, with proportionately smaller increases in baseline muscle spindle driving potentials (*Boyd et al., 1977*; *Hunt and Wilkinson, 1980*). In contrast, simulated static gamma drive primarily increased baseline muscle spindle driving potentials, and had much smaller effects on driving potentials underlying the initial burst and dynamic response during stretch (*Figure 7A–B*; *Crowe and Matthews, 1964a*; *Boyd et al., 1977*; *Crowe and Matthews, 1964b*). Even the previously reported increases and decreases in dynamic index as a result of dynamic versus static fiber stimulation, respectively, (*Figure 7C* left plot; *Crowe and Matthews, 1964a*; *Crowe and Matthews, 1964b*) were predicted by our model (*Figure 7C* right plot).

## Interactions between a biophysical muscle spindle model and muscle-tendon dynamics predict paradoxical muscle spindle firing in isometric force production

To simulate how alpha and gamma drive affect muscle spindle Ia afferent firing during voluntary movement, we simulated the peripheral mechanical interactions of the biophysical intrafusal muscle fiber above within extrafusal muscle-tendon dynamics. We modeled the muscle spindle and extrafusal muscle-tendon dynamics while the total end-to-end length was held constant to simulate isometric muscle contractions (*Dimitriou, 2014*; *Edin and Vallbo, 1990*; *Figure 8*). All the force on the tendon was generated by the extrafusal muscle fibers, as the intrafusal muscle fiber was assumed to generate negligible force. We constrained the end-to-end lengths of the intrafusal and extrafusal muscle fibers to be the same. However, muscle fibers were allowed to go slack based on their own properties, meaning the true fiber lengths were not necessarily equal when slack was present in one or more fibers. Alpha drive was simulated by activation of the extrafusal muscle fiber and gamma drive was simulated by activation of the intrafusal muscle fiber (*Figure 8A*).

We first tested the individual effects of alpha and gamma drive by independently activating extrafusal and intrafusal muscle fibers in gaussian activation patterns (*Figure 8B*, first two columns). When extrafusal muscle was activated to 50% with intrafusal activation at a nominal constant level, the stress in the extrafusal fiber increased, while the stress in the intrafusal fiber decreased (*Figure 8B*, first column, top trace, purple and green traces, respectively). The Ia afferent firing rate simulated using an integrate-and-fire neuron was abruptly silenced during extrafusal muscle contraction (*Figure 8B*, first column, middle trace); during this condition both the extrafusal and intrafusal muscle lengths decreased and tendon length increased (*Figure 8B*, First column, bottom trace, green-purple vs. orange trace). Here, the extrafusal muscle shortens due to the simulated alpha drive, and the intrafusal muscle fiber goes slack, causing a decrease in intrafusal fiber stress. In contrast, when intrafusal fiber activation (both static and dynamic) was increased and extrafusal activation was held constant, we observed no change in extrafusal stress or force, but an increase in intrafusal stress (*Figure 8B*, first column, top trace, purple and green traces, respectively). The increase in intrafusal stress causing an increased in simulated Ia afferent firing rate (*Figure 8B*, second column, middle trace) with no change in muscle fiber or tendon length (*Figure 8B*, second column, bottom trace).

When we simulated concurrent alpha and gamma drive, we observed Ia afferent firing that was dependent on the relative balance of imposed shortening of intrafusal fibers by alpha drive and the activation of intrafusal fibers by gamma drive (*Figure 8B*, third and fourth columns). As a proof of concept, we activated the intrafusal and extrafusal fibers at a 'low' and 'medium' level, respectively (*Figure 8B*, third column, top trace), which caused an increase in extrafusal muscle fiber stress, a decrease in intrafusal muscle fiber stress (*Figure 8B* third column, top trace, purple and green

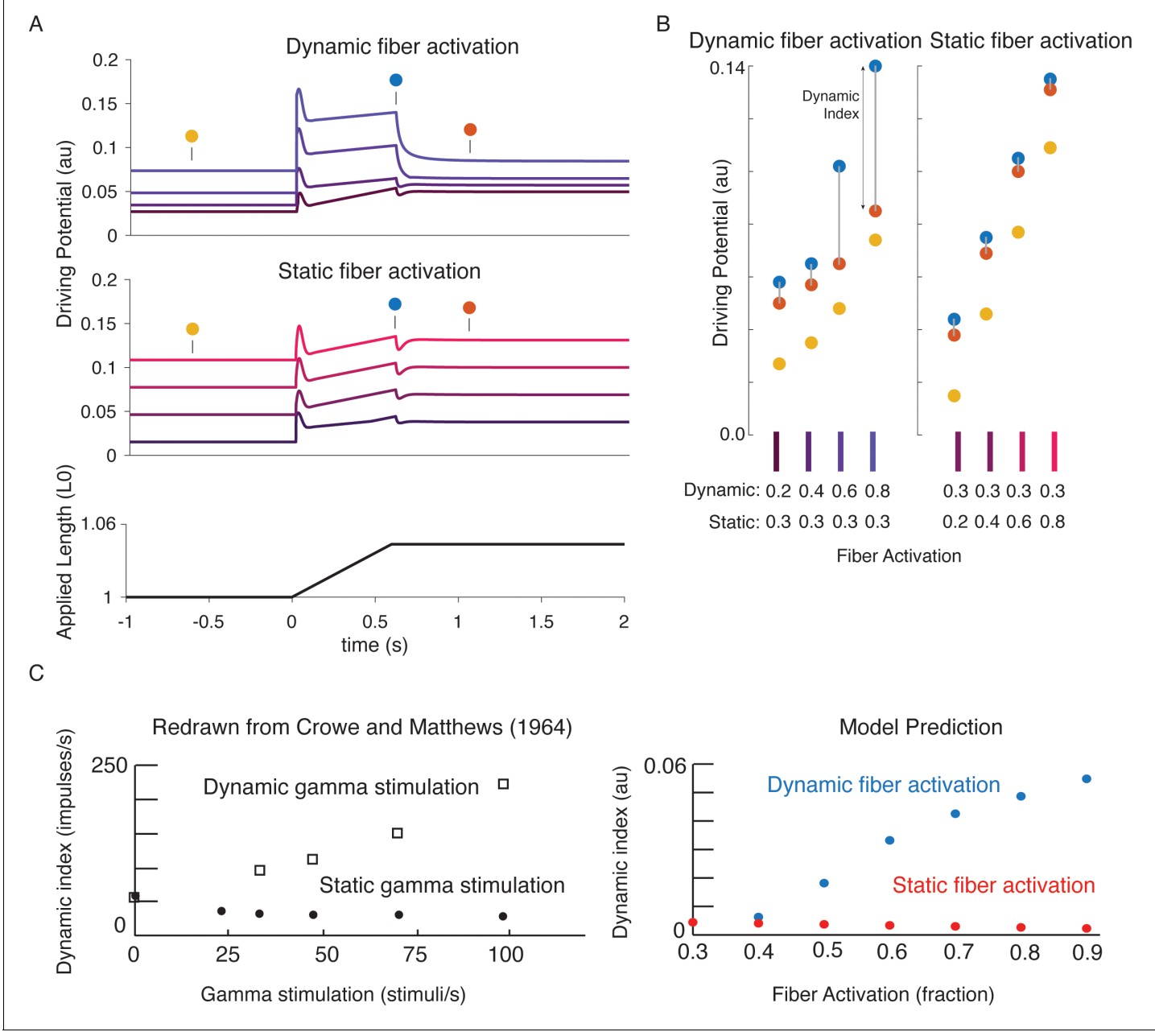

**Figure 7.** Changes in muscle spindle sensitivity caused by central drive emergent from interactions between dynamic and static fibers. Fusimotor activity was imposed on either the dynamic or the static fiber by increasing the number of active actin binding sites in the appropriate fiber. (**A**) Simulated dynamic fiber activation increased the driving potential predominantly during the ramp, with smaller increases during the background and hold period (top traces). Simulated static fiber activation predominantly increases the driving potential rate during the background and hold period, with only modest increases in during the ramp. (**B**) Emergent scaling of the dynamic index with dynamic (increase in dynamic index) and static fiber activation (decrease in dynamic index) resembled trends reported previously in the literature with (**C**) dynamic index increasing with bag fiber activation, and dynamic index decreasing with chain fiber activation, respectively. **Figure 7C** is redrawn from **Crowe and Matthews, 1964a** with permission from John Wiley and Sons.

traces), and a termination of Ia afferent firing (*Figure 8B*, third column, middle trace), while both extrafusal and intrafusal muscle fiber lengths shortened (*Figure 8B*, third column, bottom trace). Here, the extrafusal shortening due to alpha drive outweighed the effects of gamma drive to the intrafusal muscle fiber and silenced the Ia afferent (*Figure 8B*, third column). When we reversed the relative activations so intrafusal activation was 'medium' and extrafusal activation was 'low', both

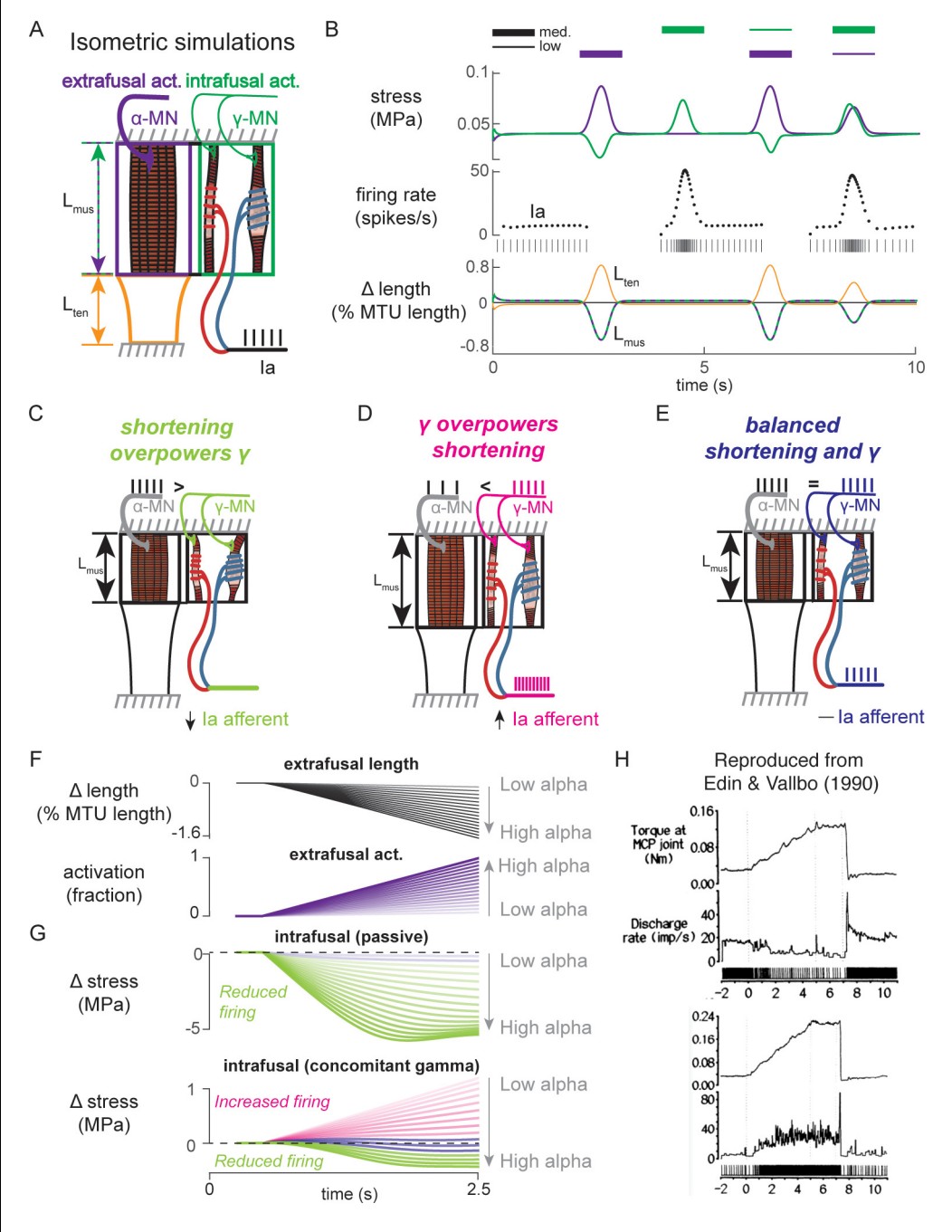

**Figure 8.** Paradoxical muscle spindle firing activity during isometric musculotendon conditions emergent from multiscale muscle mechanics. (**A**) Framework of isometric musculotendon simulations. We simulated an in-series muscle-tendon with ends fixed to its initial length. The extrafusal muscle length was shared with the intrafusal muscle fibers, which were activated independently from the extrafusal muscle. (**B**) Decoupling of intra- and extrafusal force during short isometric contractions. Different combinations of extrafusal (purple bars) and intrafusal (green bars) activation were simulated. Top row contains intra- (green) and extrafusal (purple) stress during these conditions. Middle row is the simulated Ia afferent firing rate, generated by using our mechanistic model simulated driving potential combined with a leaky integrate-and-fire neural model. Extrafusal activation shortens the extrafusal muscle, and in the absence of sufficient intrafusal activation, silences the Ia afferent firing. Intrafusal activation by itself causes the Ia afferent to fire more. When both intra- and extrafusal muscle are activated concomitantly, the relative weighting of activity determines whether the Ia afferent will increase or decrease its firing rate. (**C-E**) Schematic representation of the multi-scale mechanics responsible for Ia afferent

*Figure 8 continued on next page*

*Figure 8 continued*

behavior in these simulations. When the effects of extrafusal shortening overpower the effect of fusimotor activity on the spindle, the Ia afferent firing rate decreases, or stops altogether (left, green). When fusimotor activity overpowers the effect of extrafusal shortening on the spindle, the Ia afferent firing rate increases (center, pink). In theory, this suggests there are combinations of fusimotor activity and shortening that will result in no net change to the Ia afferent firing, despite the dynamics of the muscle (right, blue). (F) A spectrum of musculotendon isometric conditions arising from the interplay of extrafusal shortening and intrafusal activation. We activated the musculotendon model with ramp activations and simulated the intrafusal muscle force with and without concomitant gamma activation. When increasing alpha motor neuron drive (purple traces; directionally indicated in all plots by an arrow), extrafusal muscle shortened more (top plot). (G) The passive spindle always decreased its force because of the shortening imposed by the surrounding extrafusal muscle (top plot). When the intrafusal muscle was activated with concomitant drive, similar to the largest amplitude ramp of extrafusal muscle, we observed the spectrum of responses predicted in C-E: the intrafusal force either decreased (green), stayed relatively constant (purple), or increased (pink). (H) Experimental examples of one muscle spindle decreasing its firing rate (top traces) and another increasing its firing rate (bottom traces) during an isometric task in human. *Figure 8H* is reproduced from *Edin and Vallbo, 1990* with permission from The American Physiological Society. It is not covered by the CC-BY 4.0 license and further reproduction of these panels would need permission from the copyright holder.

intrafusal and extrafusal muscle fiber stress was increased while the muscle shortened (*Figure 8B*, fourth column, top versus bottom traces), causing increased Ia afferent firing (*Figure 8A*, fourth column, middle trace). In this case, both extrafusal and intrafusal muscle fibers were shortening, but the presence of adequate intrafusal activation maintains the force in the intrafusal muscle fibers, driving the muscle spindle firing in the presence of alpha drive to the extrafusal muscle fiber (*Figure 8B*, fourth column).

The simulations just described illustrate the multiscale, complex, interactions between the musculotendon and the muscle spindle in a few intuitive cases (*Figure 8C–E*). Briefly, if the extrafusal muscle shortening caused by alpha motor drive is not sufficiently counteracted by gamma drive, the Ia afferent will reduce its firing rate or cease firing altogether (*Figure 8C*). Alternatively, if the gamma drive to the spindle stretches the sensory region at a greater rate than the extrafusal shortening, the Ia afferent will increase its firing rate (*Figure 8D*). Finally, it follows that there is a set of extra- and intrafusal states that will lead to no change in Ia firing rate during isometric conditions (*Figure 8E*).

We were inspired to use these observations to explain paradoxical Ia afferent firing rates from experiments in human finger muscles (*Figure 8C*; *Edin and Vallbo, 1990*), in which muscle spindle Ia afferents were shown to either increase or decrease their firing rates (*Figure 8C*, bottom traces) during an isometric contraction despite the same joint torque profile (*Figure 8C*, top traces). We mimicked muscle fiber shortening under isometric conditions by applying a ramp increase in extrafusal muscle activation (*Figure 8D*). As expected, increasing alpha drive decreased extrafusal muscle fiber length (*Figure 8B*, black trace), and increased extrafusal muscle fiber stress (*Figure 8D*, purple trace). In the absence of gamma drive, the intrafusal fiber stress decreased as alpha drive increased (*Figure 8D*, green trace) resulting in no muscle spindle firing as shown experimentally (*Elek et al., 1990*). When we increased gamma drive by a fixed amount intrafusal fiber stress could increase (*Figure 8D*, pink traces), decrease (*Figure 8D*, green traces), or experience no change (*Figure 8D*, purple traces) in stress depending on the amount of alpha drive. As such it was possible to generate almost any level of muscle spindle firing during isometric conditions where the muscle fibers shorten by varying the relative levels of alpha and gamma drive.

Finally, to demonstrate the complex interplay of alpha and gamma drive when muscle fibers are lengthened, we applied ramp-and-hold stretches to the model musculotendon while simulating the Ia afferent response under different extra- and intrafusal activation conditions. When the musculotendon was passive, i.e. no alpha drive, the muscle fiber length closely followed the applied musculotendon length (*Figure 9A*, left panel). Conversely, when the extrafusal muscle fiber was activated during the stretch, there was a marked difference in musculotendon length compared to the applied length (*Figure 9A*, right panel). For each of these cases, we simulated muscle spindle Ia afferent responses with and without gamma drive. In the case of the passive spindle, the musculotendon stretch in the absence of alpha drive produced a slightly higher muscle spindle Ia afferent response than in the presence of alpha drive (*Figure 9B*, top row). Specifically, the baseline firing rate, initial

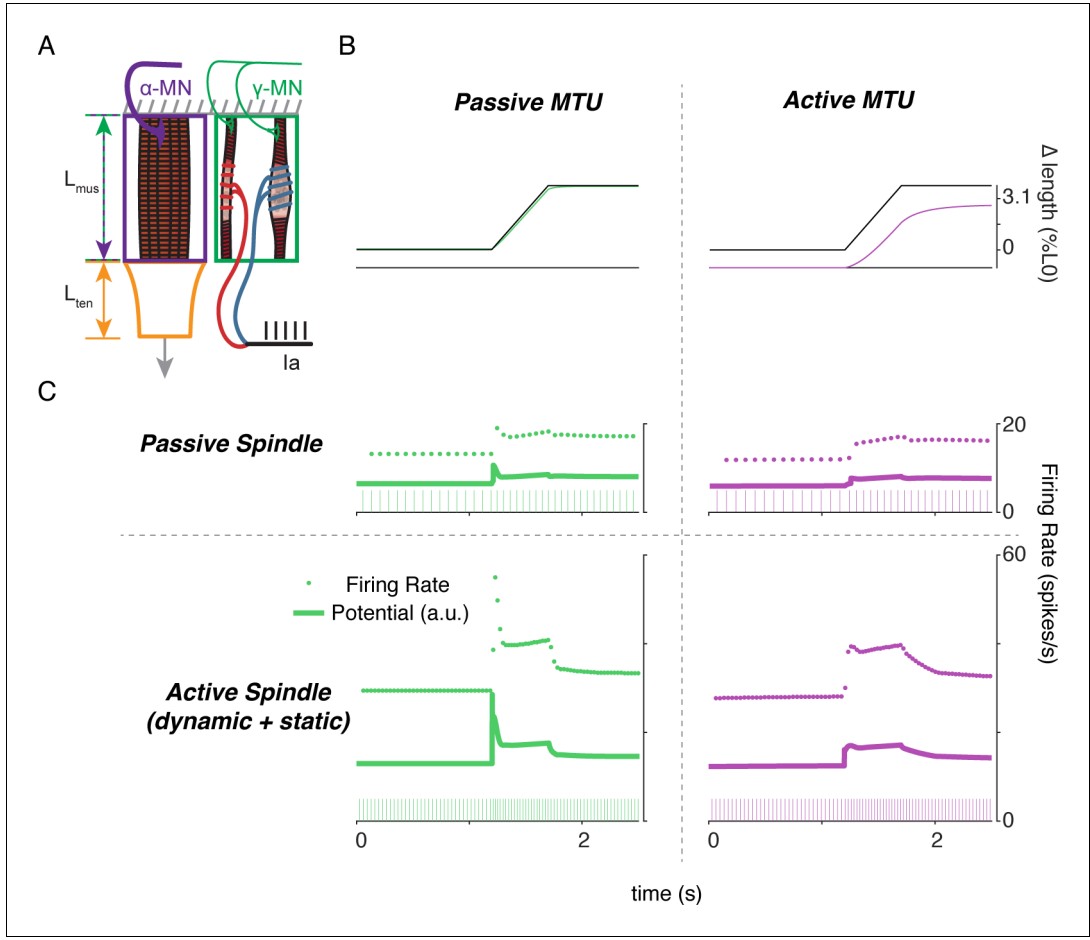

**Figure 9.** Complex interplay of movement, alpha, and fusimotor drive underlie muscle spindle Ia firing rates. (**A**) Framework of free-end musculotendon simulations. We simulated an in-series muscle-tendon with one end fixed and one end free to move. The extrafusal muscle length was shared with the intrafusal muscle fibers, which were activated independently from the extrafusal muscle. (**B**) Passive musculotendon has a relatively compliant muscle, leading to a MTU length change being primarily applied to the muscle, not the tendon. Extrafusal activation creates a relatively stiff muscle, leading to a smaller length change for the same MTU length applied. (**C**) Musculotendon activation changes the coding properties of both passive and active muscle spindles. When the MTU is passive, a passive muscle spindle has an initial burst.

burst, and dynamic response were all slightly larger in the case of the passive musculotendon. These differences were more exaggerated when we simulated gamma drive by increasing intrafusal muscle fiber activation (*Figure 9B*, bottom row). The addition of a tonic level of gamma drive increased the baseline firing rate, the amplitude of the initial burst, and the dynamic response to stretch in the passive extrafusal musculotendon unit (*Figure 9B* bottom row, green trace). Adding alpha drive decreased baseline firing only slightly, abolished the initial burst, and had little effect on the dynamic response to stretch (*Figure 9B* bottom row, purple trace). These simulations demonstrate the importance of considering musculotendon dynamics and activation when estimating muscle spindle afferent feedback in different experimental conditions.

## Discussion

### Summary

Here, we present a multiscale biophysical model that predicts a wide range of muscle spindle Ia afferent firing characteristics in both passive and active conditions, illustrating the complex central and peripheral neuromechanical interactions that underlie them in naturalistic conditions. We

demonstrated that a diverse range of Ia afferent firing characteristics during passive stretch – well-documented in the literature but not completely understood mechanistically – is emergent from first principles of intrafusal muscle contractile (cross-bridge) mechanisms. Ia afferent firing in a broad range of behaviorally relevant conditions can be reproduced by converting the force and yank of the intrafusal muscle fibers within the muscle spindle into afferent driving potentials using a simple linear relationship. This transformation represents the yet unexplained mechanotransduction process in the equatorial encoding region of muscle spindle sensory organs. The multiscale mechanical interaction of the muscle spindle within the muscle tendon unit revealed isometric force generating conditions in which muscle spindle firing followed intrafusal muscle fiber force, which can either resemble or differ from muscle-tendon unit force or length depending on the relative amounts of alpha and gamma motor neuron drive. As such, mechanical interactions within the muscle-tendon unit and forces external to the muscle all contribute to firing rates of muscle spindle Ia afferents; they may even create firing patterns considered to be paradoxical if viewing muscle spindles as simple passive sensors of kinematic or kinetic variables. Finally, we simulated multiscale mechanical interactions including movements, alpha activation of extrafusal muscle, and gamma activation of intrafusal muscle to demonstrate the importance of these complex interactions when modeling proprioceptive activity. Our neuromechanical framework demonstrates how muscle spindle firing can reflect the complex interplay of external and self-generated forces and motion. Importantly, our model provides a foundation for biophysically realistic, predictive muscle spindle models that can be extended to understand how properties of tissues, muscles, neurons, and central drive affect muscle spindle sensory function in health and disease.

## A biophysical muscle model connects many independent observations with a simple hypothesis

Our approach of starting with biophysical principles of muscle force generation enabled us to develop a muscle spindle model that predicts nonlinear muscle spindle firing properties beyond the conditions that it was designed to replicate, without any change in model parameters. We simulated muscle cross-bridge kinetics to produce the muscle fiber force and yank that drive history-dependent responses to a repeated muscle stretch stimulus, including the initial burst. Other muscle spindle models do not exhibit history-dependence and – by design – either lack (*Mileusnic et al., 2006*) or always generate (*Hasan, 1983*) an initial burst at the onset of stretch. In our model, the initial burst emerges from the number of cross-bridges that are formed after a period of rest, temporarily increasing the stiffness of the fibers before being stretched to their mechanical limit. This number depends on the prior and current fiber length, velocity, acceleration, and activation of the intrafusal fibers, and thus leads to the modulation of the initial burst of the spindle afferent with these variables. Similarly, the fractional power relationship between muscle spindle dynamic responses and stretch velocity (*Hasan, 1983*; *Lin and Crago, 2002*) has previously been imposed by including it explicitly within the model definition. While it can be simulated using a phenomenological muscle model with a built-in fractional power function, prior models fail to account for the history dependence of the velocity sensitivity and thus fail to replicate muscle spindle firing patterns outside of tightly controlled laboratory conditions. Similarly, the adaptation of muscle spindle firing rates when held at a constant length after stretch follows the decrease in force of the intrafusal muscle fiber when it is held at a constant length (*Blum et al., 2017*; *Husmark and Ottoson, 1971*). Here, history dependence, the nonlinear relationship to stretch velocity, the scaling of the initial burst to acceleration, and firing rate adaptation are all implicit, emergent properties of intrafusal cross-bridge force and yank under different conditions – no explicit formulation of these relationships is necessary.

Even the apparent differential effects on Ia afferent encoding by static and dynamic fusimotor activity are accounted for by the mechanics of muscle force generation (*Crowe and Matthews, 1964a*; *Crowe and Matthews, 1964b*). We designed our model with slow cycling myosin in the 'dynamic' fiber and fast cycling myosin in the 'static' fiber, modeled after nuclear bag and chain fibers, respectively (*Thornell et al., 2015*). In doing so, the dynamic and static fusimotor effects on Ia afferent stretch sensitivity were closely predicted, emerging from muscle biophysics and a simple representation of mechanotransduction (*Banks et al., 1997*). By taking a structural approach to the interactions between intrafusal and extrafusal muscle fibers the effect of gamma motor drive were more robustly simulated (*Lin and Crago, 2002*).

Variations in force and yank sensitivity account for both overall firing characteristics as well as 'static-dynamic' variability across Ia afferents during passive stretch. The few available recordings of muscle spindle receptor potentials during muscle stretch exhibit fast initial transients similar to muscle fiber yank, as well as a sustained potential that resembles muscle force and its decay following a stretch (*Hunt and Ottoson, 1975*). We show that sweeping a range of sensitivities to intrafusal force and yank predicts a continuum of firing phenotypes resembling those seen previously in both healthy and perturbed conditions (*Housley et al., 2020b*; *Vincent et al., 2016*). Further, each set of sensitivities can predict muscle spindle firing across different stretch velocities and displacements. Whether a similar principle describes the firing of the second type of muscle spindle afferent, group II afferents, remains to be tested.

## Multiscale interactions are necessary to consider for natural behavior

Explicit simulation of the mechanical arrangement of the intrafusal muscle fibers within the muscle-tendon unit enabled us to demonstrate the profound effects that central drive to alpha and gamma motor neurons can have on muscle spindle firing during active force and muscle stretch conditions. Our work emphasizes the importance of modeling the anatomical arrangement of the muscle shown in some prior models (*Lin and Crago, 2002*; *Mileusnic et al., 2006*). With the addition of biophysical force generation mechanics we now provide a computational testbed to better explore the long-discussed idea that muscle spindle firing rates are driven by properties of intrafusal muscle fiber forces (*Matthews, 1981*), and determined by both the central drive to intrafusal and extrafusal muscle fibers, as well as interactions with external forces on the muscle. When simulating the mechanical interaction of the muscle spindles within the muscle tendon unit, our model reveals that muscle spindle firing can depend critically on the activation, force, and length of the intrafusal fiber within the muscle spindle, which shapes the degree to which muscle spindle firing resembles extrafusal muscle length, velocity, force or other biomechanical variables (*Bewick and Banks, 2015*; *Kruse and Poppele, 1991*).

The multiscale model presented here provides a mechanistic framework that can explain the rich diversity of movement-related biomechanical signals in naturalistic behaviors. There are many such scenarios we have not simulated that our model may provide insights for. For example, the mechanical interactions between intrafusal and extrafusal muscle fibers (*Figures 8* and *9*) may explain the complex relationships between Ia afferent signals in locomotion to muscle length and force. During the swing phase of the cat locomotor step cycle, relaxed ankle extensor muscles are stretched by activity in ankle flexor muscles, and extensor Ia firing rates appear to follow muscle velocity and/or length (*Prochazka and Gorassini, 1998a*; *Prochazka and Gorassini, 1998b*), which closely resemble extrafusal and intrafusal muscle force and yank (*Blum et al., 2017*) in passive conditions. However, the same muscle spindle Ia afferent also fires in early stance when there is no change in muscle length, but the muscle is both active due to alpha drive, and loaded due to carrying the weight of the animal: this firing is likely due to intrafusal force resulting from the interactions between gamma drive, alpha drive, and the external load (*Figure 8*, green traces). The same muscle spindle afferent ceases firing in midstance, when the intrafusal muscle is possibly activated by gamma drive, but not enough to prevent the muscle spindle from falling slack as the extrafusal muscle fiber shortens due to alpha drive (*Ellaway et al., 2015*; *Prochazka, 2015*; *Prochazka and Gorassini, 1998b*; *Figure 8*, purple traces). We did not simulate such experiments, but doing so would provide an excellent opportunity to further improve our framework for natural movements.

Complex biomechanical interactions may further explain muscle spindle afferent activity that have heretofore proven challenging to model and to understand mechanistically. Ia afferent firing has been shown in humans to be highly dependent on resistive loads applied to an attempted movement (*Dimitriou, 2014*; *Edin and Vallbo, 1990*; *Vallbo, 1974*). An interpretation of this observation could be that, in coupled agonist-antagonist systems, Ia afferents under alpha-gamma coactivation or beta motor control signal the effects of loads on a joint originating outside their parent muscles. For example, in an unimpeded concentric contraction, the antagonist Ia afferent stretches and signals the effect on the joint torque and angle caused by the agonist contraction. If, in a similar scenario, a load prevents the agonist from shortening, there would be no change in the antagonist Ia afferent activity, correctly signaling the lack of joint movement. Simultaneously, the agonist Ia afferents would increase their firing due to concomitant fusimotor drive, signaling the impeding effects of the external load on the joint. In both cases, the Ia afferent, whether in an agonist or antagonist

muscle, would fire when interacting with a force that originates outside the parent muscle. Whether the antagonist load or reciprocal inhibition of the fusimotor drive between antagonist muscle pairs explains these results, Ia afferents in opposing muscles may work together to signal joint-level information about external loads (*Dimitriou, 2014*). Simulating such intermuscular interactions is likely necessary to predict proprioceptor activity during movement, and would require multiple muscle and muscle spindle models to be coupled.

## Limitations of our current model and future directions

The core framework presented provides proof of concept for more elaborate and biophysically accurate model of muscle spindle firing necessary to predict muscle spindle signals during naturalistic movement conditions in health and disease. Importantly, our model can be used as a platform to enable the development of computational models capable of testing the effects of a wide variety of multiscale mechanisms on muscle spindle firing, including architectural arrangement of the muscle spindle within the muscle (*Maas et al., 2009*), intra- and extrafusal muscle myosin expression, more complex muscle force-generating mechanisms, extracellular matrix stiffness, mechanosensory encoding mechanisms, and biophysical neural dynamics that could all be affected by aging and disease.

Our primary focus in developing our mechanistic model was to use the simplest two-state cross-bridge model of actin-myosin interactions to qualitatively, but not quantitatively, reproduce history dependence in its stretch responses (*Figure 6A*) as seen in biological Ia afferents (*Figure 2B*). It should be noted that, despite the discrepancies in the more abrupt stretch-shorten 'triangle' stretches (*Figure 6A*), our model was able to closely reproduce the human Ia afferent response (*Figure 6B,C*) during the slow sinusoidal stretch, which may be more physiologically relevant. However, we acknowledge the apparent shortcomings of our chosen model in reproducing the triangle responses from anesthetized rat. With our approach, it is impossible to separate the elastic contributions from endomysial tissue from those in myofilaments, from structures such as titin. It remains to be seen which aspects of complex muscle spindle properties will improve the model's predictive capabilities, but it may be necessary to account for muscle-specific (*Campbell, 2014*) kinetic schemes or to replace the simple linear model of titin with more sophisticated properties (*Nishikawa et al., 2012*) to capture calcium-dependent hysteresis in regulation of titin stiffness and a range of other complex processes of muscle force generation (*Campbell, 2016*; *Campbell, 2017*; *Mann et al., 2020*). We relied on the similarly timed short-range elastic component of extrafusal force response to predict the putative intrafusal mechanics-driven Ia afferent stretch responses. Additional sources of history-dependence could include history dependence in extrafusal muscle fibers and non-contractile tissues, or the inertial of the muscle mass when rapidly accelerated. Another possibility is that we failed to account for the complex process(es) responsible for the occlusion between multiple transduction zones in the primary afferent (*Banks et al., 1997*; *Mileusnic et al., 2006*), in which the firing rate may be more dependent on the bag1 fiber during the initial stretch and become more dependent on to bag2 and chain fibers in later stretches.

A comprehensive understanding of the mechanisms underlying muscle spindle mechanotransduction, that is, translation of mechanical stimuli to receptor potentials, remains elusive. Lacking sufficiently detailed information from the literature, we represented the entire neuromechanical transduction process by a set of constant gains. However, we do recognize that the dynamics of yet uncharacterized viscoelastic properties of the equatorial regions of intrafusal muscle fibers, ion channels in muscle spindle afferent endings (*Bewick and Banks, 2015*; *Carrasco et al., 2017*), and intrinsic neural dynamics (e.g. those underlying neural history dependence, spike-frequency adaptation, and occlusion *Banks et al., 1997*) of the afferent plays a significant role in the resulting muscle spindle firing signals. Finally, we focused only on the muscle spindle Ia afferent, but the anatomical arrangement of our model could also provide a biophysical framework for understanding muscle spindle group II afferents that are located at the junction between the polar and equatorial regions of intrafusal muscle fibers, as well as and Golgi tendon organ Ib afferent located at the extrafusal musculotendinous junction. In future studies, it would also be beneficial to adopt the approach of *Mileusnic et al., 2006* and optimize the parameters of our model to match the careful experimental conditions of specific muscles and animal models.

## Conclusion

Ultimately, it is clear that the complex interplay between central drive, internal, and environmental forces cannot be ignored when decoding muscle spindle firing patterns, especially in voluntary movement and during interactions with the environment. Such neuromechanical interactions are key in interpreting the afferent signals and percepts in a variety of conditions where mismatches in extra-fusal and intrafusal mechanical state due to fatigue or pre-conditioning alter the perception of both force and position (*Proske et al., 2014*; *Proske and Gandevia, 2012*), or the role of muscle spindles in perception of force and weight (*Jami, 1992*; *Luu et al., 2011*; *Savage et al., 2015*). Our multi-scale biophysical model provides a new biophysical framework by which we may move beyond the idea that muscle spindles can be characterized as simple passive encoders of biomechanical signals during movement, enabling a more sophisticated understanding of muscle spindles as proprioceptive sensors, their role in neural control of movement, and how neurological disorders disrupt sensorimotor systems.

# Materials and methods

## Animal care

All procedures and experiments were approved by the Georgia Institute of Technology's Institutional Animal Care and Use Committee (protocol A16038). Adult female Wistar rats (250–300 g) were studied in terminal experiments only and were not subject to any other experimental procedures. All animals were housed in clean cages and provided food and water ad libitum in a temperature- and light-controlled environment in Georgia Institute of Technology's Animal facility.

## Terminal physiological experiments

Experiments were designed to measure the firing of individual muscle afferents in response to controlled muscle stretch using conventional electromechanical methods applied in vivo as reported by *Vincent et al., 2016* (see *Figure 1*). Briefly described, rats were deeply anesthetized (complete absence of withdrawal reflex) by inhalation of isoflurane, initially in an induction chamber (5% in 100% $O_2$) and maintained for the remainder of the experiment via a tracheal cannula (1.5–2.5% in 100% $O_2$). Respiratory and pulse rates, $Pco_2$, $P_{O_2}$, blood pressure, and core temperature were maintained within physiological ranges variously by injecting lactated ringers subcutaneously and by adjusting anesthesia and radiant heat. Rats were secured in a rigid frame at the snout, lumbosacral vertebrae, and distally on the left femur and tibia (knee angle 120°). Dorsal roots S1-L4 were exposed by laminectomy and supported in continuity on a bipolar recording electrode. Triceps-surae muscles in the left hindlimb were dissected free of surrounding tissue and their tendon of insertion was cut and attached to the lever arm of a force and length-sensing servomotor (Model 305C-LR, Aurora Scientific Inc). Triceps surae nerves were placed in continuity on monopolar stimulating electrodes, and all other nerves in the left hindlimb were crushed to eliminate their contribution to muscle-stretch evoked action potentials recorded in dorsal roots. Exposed tissues were covered in pools of warm mineral oil Surgical and recording preparation followed by data collection (see below) lasted for up to 10 hr. At the conclusion of data collection, rats were killed by exsanguination preceded either by overdose with isoflurane inhalation (5%).

Data collection centered on trains of action potentials recorded from individual primary sensory neurons responding to length perturbations of the parent triceps surae muscles. Action potentials were recorded intracellularly from axons penetrated by glass micropipettes in dorsal root. Sensory axons were identified as triceps surae when responding with orthodromic action potentials evoked by electrical stimulation of the corresponding muscle nerves. Axons selected for further study were classified as group Ia when (a) isometric twitch contractions interrupted muscle stretch-evoked firing, (b) each cycle of high frequency (>200 Hz), low amplitude (80 µm) muscle vibration evoked action potentials, and (c) muscle stretch produced an initial burst firing of high frequency (>100 Hz) firing at the onset of rapid muscle stretch.

Ia afferents were studied for their firing responses to two forms of muscle stretch beginning from and returning to $L_{r(rest)}$o, which was the muscle length corresponding to 10 grams passive muscle force: (1) ramp-hold-release stretch (20 mm/s constant velocity ramp and release with intervening hold to 3 mm length) repeated every four secs and (2) sequential triangular ramp-release stretches

to 3 mm at constant velocity (4 mm/s). Records of action potentials, muscle force and length were digitized and stored on computer for later analysis.

In the 11 Ia afferents for which the initial pseudolinear model analyses were performed, a range of 6–99 stretch trials with varying maximum length and velocity were achieved depending on the recording stability. To ensure sufficient information for statistical measures, we required that stretch trials have at least 50 recorded action potentials in order to be included in statistical analyses. Stretch trials where spikes were not discriminable were excluded. These criteria yielded suitable datasets for 11 individual afferents from five animals for pseudolinear model analyses and six individual afferents from five animals for the axonal stimulation analyses. We also included three afferents from three rats treated with oxaliplatin (*Bullinger et al., 2011*).

## Muscle fiber force estimation

To isolate the component of recorded musculotendon force arising from the muscle fibers (used as a proxy for intrafusal muscle force), we assumed an idealized musculotendon mechanical arrangement (*Figure 1*). In summary, we assumed there was noncontractile passive connective tissue arranged mechanically in parallel with the muscle fibers and removed it analytically, as previously described (*Blum et al., 2019*).

Briefly, we assumed the noncontractile tissue acted as a nonlinear spring of the form:

$$F_{nc} = k_{lin}(L - L_0) + Ae^{k_{exp}(L - L_0)}$$

where $k_{lin}$, $k_{exp}$, and $A$ are greater than or equal to zero. Once parameters were selected by the optimization procedure, the estimated noncontractile tissue forces were subtracted from the recorded force to estimate the muscle fiber force, which was fit to the IFRs.

## Pseudolinear models for predicting firing responses

We predicted spiking responses using pseudolinear combinations of either recorded musculotendon length-related (length, velocity, and acceleration) or force-related (estimated muscle fiber force and yank) variables (*Figure 2—figure supplement 1*). The relative weights and offsets for each variable in a model were optimized to minimize the squared error between the model prediction and Ia spike rates on a per-trial basis.

For both the force- and length-related models, we fit the estimated IFR for each model to the IFR of the afferent for each stretch trial included in our analyses (for all 20 afferents presented in this study). The model parameters, consisting of a weight ($k_i$) and offset ($b_i$) for each force- or length-related variable included in the sum, were found via least-squares regression using MATLAB's optimization toolbox (*fmincon.m*) and custom scripts.

We observed a peak to peak delay from the whole musculotendon yank and the initial burst, likely caused by delayed force transmission to the intrafusal fibers from the tendon (*Blum et al., 2019*). A time delay ($\lambda_j$) was determined by shifting the timestamp of the variables forward relative to the IFR data to be fit (note: this time delay was 0 for all variables except yank, to account for the apparent delay between the onset of muscle force response and the onset of the spiking response). The general form of the models was:

$$IFR_{j,n}(t) = \left( \sum_{i=1}^{n} k_i \cdot \left( x_i(t - \lambda_j) + b_i \right) \right)$$

where the IFR estimate of the *j*th model for the nth perturbation was estimated by a sum of *n* force- or length-related variables, offset by a single value, $b_i$, and scaled by a gain, $k_i$. $\lfloor \rfloor$ denote positive values of the argument. Model estimates for IFR were related to the recorded IFR of the *m*th afferent by the equation:

$$IFR_{j,n}(t) + e(t) = IFR_{m,n}(t)$$

Error, $e(t)$, was minimized by finding the set of parameters for each model that minimizes a measure related to $e(t)^2$.

### Antidromic axonal stimulation dataset

To test whether the force and yank components could arise from separate mechanosensitive mechanisms, another set of experiments was performed on six additional afferents in four animals. Each dataset consisted of three ramp-hold-release stretches repeated at 4 s intervals. The second of three stretches was preceded by antidromic action potentials in a 50 Hz train lasting 500 ms. These action potentials were isolated exclusively to the Ia afferent under study by injecting suprathreshold current pulses (0.5 ms) directly into the axon through the micropipette located in the dorsal root. The first and third stretches were unconditioned and served as bookend controls.

For each trial in these six afferents, we found the best-fit prediction for the force-related model using the parameter optimization described earlier. For the pre- and post-stimulation control trials, we first fit the model without a yank component, and then refit the model with a yank component. For the trials in which the electrical stimulus was applied, the yank component was set to be zero and the force and constant components were optimized as described before.

We performed one-way ANOVA on model performance ($R^2$), yank sensitivity ($k_Y$), force sensitivity ($k_F$), and the constant component (C) across 5 groups of model fits: pre-stimulus control trials without (1) and with (2) yank sensitivity, stimulus trials (3), and post-stimulus trials without (4) and with (5) yank sensitivity. We used the Tukey-Kramer method to examine all pairwise comparisons between groups.

### Oxaliplatin dataset

We used data collected previously to test whether force and yank components were altered by oxaliplatin chemotherapy alone (*Bullinger et al., 2011*). The effect of oxaliplatin on sensory coding of Ia afferents has been well-documented, so we analyzed three afferents from different animals. We fit the muscle fiber force-related model (described above) to three stretch trials for each afferent (3 mm, 20 mm/s). We performed one-way ANOVA on model performance ($R^2$) between model fits with and without yank for each afferent to test the significance of the yank component on model performance.

### Applying estimated fiber force-related driving potentials to model neuron

To test the feasibility of the force, yank, and constant components of the muscle fiber force-related model as mechanical signals encoded by the muscle spindle receptor, we applied a range of combinations of components to a conductance-based model neuron (based on the Connor-Stevens model; see next section) and examined the resulting firing rates. We first estimated the muscle fiber force and yank, as described previously, and varied the relative gains of these signals before adding them with a constant component. Once the components were added together, they were half-wave rectified, and applied to the model neuron as a stimulus current.

Model neuron sensitivities to these components were tuned until the model instantaneous firing rate was within 10 spikes/s for initial burst, dynamic response, and final plateau. We treated the parameter values which produced this response as the nominal values for the model. The relative sensitivities of the model neuron to force and yank component were then swept from 10 to 600% of their respective nominal values. We then compared the resulting changes in predicted firing rates with different phenotypical muscle spindle responses observed from these and other experiments.

### Responses of muscle spindle ia afferents to stretch

Consistent with prior studies, all Ia afferents exhibited initial bursts at onset of applied stretch, followed by a dynamic response during constant velocity stretch, and a period of rate adaptation during the subsequent isometric hold period. When repeated ramp-release length-changes were applied to the muscle, an initial burst and dynamic response was present during the first ramp, but the initial burst was absent and dynamic response was reduced during subsequent stretches–a phenomenon in Ia afferents known as history-dependence (*Haftel et al., 2004*).

The population of 11 Ia afferents considered for the first analysis varied in sensitivity to stretch length, velocity, and acceleration. More dynamic afferents, as quantified by dynamic index (*Matthews, 1963*) typically had relatively large spike responses during positive velocity stretch. More static afferents exhibited more firing during the plateau phase of stretch, with relatively smaller

dynamic indices. The population of afferents also exhibited a range of initial burst amplitudes in response to stretch. There was no clear relationship between the dynamic index and initial burst amplitudes for a given afferent. Despite the differences in sensitivity amongst the afferent population, the waveforms of afferent responses to the same stretch stimuli contained the same features (i.e. all afferents exhibited initial bursts, dynamic responses, and rate adaptation to varying degrees).

## Conductance-based model neuron for reproducing spiking activity

To demonstrate the plausibility of force- and yank-related ionic currents caused by stretch, we used a modified Connor-Stevens conductance-based model neuron to model the transformation of graded receptor potentials into action potentials by the afferent (*Connor and Stevens, 1971*). The model neuron contained a fast sodium, delayed rectifier potassium, transient potassium, and leak conductances implemented in Simulink using built-in differential equation solvers (ode23s.m).

## Intrafusal muscle model

All intrafusal muscle model code and scripts used to generate the data presented in this study are available at https://github.com/kylepblum/MechanisticSpindleManuscript.git (copy archived at swh: 1:rev:da0ed89078a948167b4e2b511480787ddb681892. If interested in using the model as implemented in this study, a description of how to use it is also available there (*Blum and Campbell, 2020*).

   To simulate the hypothesized history-dependent mechanisms of intrafusal muscle fibers, we used a computational model of cross-bridge cycling. We implemented a model in MATLAB based on a simplified structure of the model developed by Campbell (*Campbell and Lakie, 1998*; *Campbell and Moss, 2000*; *Campbell, 2014*; *Campbell and Moss, 2002*; *Figure 4—figure supplement 1*). Instead of simulating the coupled dynamics between myosin heads and actin binding sites (*Campbell, 2014*), we focused on the thick filament kinetics and greatly simplified the thin filament. In brief, the force in each half sarcomere was calculated as a sum of two components: an active component generated by the cycling activity of a population of myosin heads and a passive component generated by a simulated linear spring modelling the contributions of titin in the half sarcomere:

$$Total\ Force = \int_{-\infty}^{+\infty} \left( k_{cb}\ \rho\ f(x)\ (x + x_{ps})\ dx \right) + k_{pas}(l_{hs} - l_0)$$

   The model calculates the force of a half sarcomere at each time step by adding the forces generated from each myosin head attached to an actin binding site (active force) with the elastic force of titin (passive force). The active force is calculated as the fraction of attached myosin heads, $f$, multiplied by the number density, $\rho$, unit stiffness of a single attached actin-myosin cross-bridge, $k_{cb}$, multiplied by the length of the cross-bridges, $x + x_{ps}$ (where $x_{ps}$ represents the additional displacement of an attached cross-bridge required to generate a 'powerstroke' force when multiplied by $k_{cb}$), integrated across cross-bridge lengths. The passive force is calculated as the length of the half sarcomere, $l_{hs}$, relative to a reference length, $l_0$, multiplied by a linear stiffness, $k_{pas}$.

   To simplify the coupled dynamics of Campbell, we decided to control intrafusal activation directly instead of ionic calcium concentration:

$$\frac{df_{act}(t)}{dt} = u(t),$$

   Where the change in the fraction of activated actin binding sites, $df_{act}(t)$, is determined by user input u(t).

   The myosin cycling dynamics were modelled using a two-state system in which each myosin head could either be attached as a cross-bridge with length $x$, or detached. The numbers of myosin heads in each of these states were governed by the system of partial differential equations, as in Myosim:

$$\frac{\delta A(x,t)}{\delta t} = k_f(x)D(t) - k_g(x)A(x,t)$$

$$\frac{\delta D(t)}{\delta t} = \int\limits_{-\infty}^{+\infty} k_g(x)A(x,t)dx - \int\limits_{-\infty}^{+\infty} k_f(x)D(t)dx$$

where $A(x,t)$ is the number of attached myosin heads attached to actin binding sites at time $t$ extended by length $x$ and $D(t)$ is the number of myosin heads in the detached state. The rate equations $k_f(x)$ and $k_g(x)$ are the forward and reverse rates of myosin attachment, respectively, and are both functions of cross-bridge length, $x$. The partial differential equations were simplified into a system of ODEs by solving the time-dependent equations simultaneously at each cross-bridge length. The resulting system of ODEs was solved using a built-in Matlab ODE solver (*ode23*.m). Thick filament dynamics are coupled to the activation of the thin filament by the number of cross-bridges participating in cycling:

$$n(t) = \int\limits_{-\infty}^{+\infty} A(x,t)dx + D(t)$$

where the fraction of bound cross-bridges is defined by the thick filament:

$$f_{bound}(t) = \int\limits_{-\infty}^{+\infty} A(x,t)dx,$$

and the number of detached cross-bridges is defined by the thin filament activation and $f_{bound}(t)$:

$$D(t) = f_{act}(t) - f_{bound}(t).$$

There are two modes of operation for these muscle models: length control model and slack mode. The model switches between the two modes automatically, based on whether each intrafusal fiber is taut (force greater than zero) or slack (force equal to zero).

Length control mode is used by the model when the half sarcomere length is equal to the end-to-end length of the sarcomere (i.e., the half sarcomere is not slack). When in length control mode, the command length is applied directly to the half sarcomere, and the system of differential equations is solved as described earlier. The half sarcomere is updated in the following order: (1) The length of the half sarcomere and calcium concentration are each updated by the change in command length and change in command calcium concentration applied to the model, respectively, (2) the amount of filament overlap is updated, (3) thin filament dynamics are updated, (4) the population of myosin heads are allowed to evolve, and (5) the population of myosin heads is moved according to the change in length of the half sarcomere.

Slack mode contains several extra steps for solving the equations governing the model. At every time step, the model uses an iterative search (*fzero.m*) to find the length at which the force in the current-state half sarcomere would reach zero and compare it to the command length. If the command length is smaller than the 'slack length' calculated, the model enters into slack mode; otherwise, the model stays in length control mode.

Once the model enters into slack mode, the model first evolves the cross-bridge distribution in time based on the system of differential equations described earlier. The model then repeats the iterative search to find the slack length for the half sarcomere given the new cross-bridge distribution and shifts it to the slack length for the half sarcomere. At the next time step, the model repeats the comparison between the command length and the slack length for the current state of the half sarcomere. If the command length is greater than the slack length, the model returns to length control mode; otherwise, the model stays in slack mode.

## Intrafusal muscle model parameters

Model parameters were either chosen to match the default parameters from *Campbell, 2014*, so the model would exhibit history-dependence at the time-course measured in this work, or based on the limited qualitative information we have regarding intrafusal muscle fibers (*Thornell et al., 2015*). All simulations used the same set of model parameters (*Supplementary file 1*).

Myosin attachment and detachment rates equations, $k_f(x)$ and $k_g(x)$, were selected such that the force response of the model would exhibit history-dependent features consistent with observations in both permeabilized muscle fibers and instantaneous firing rates of muscle spindle Ia afferents. The bag and chain fibers were differentiated by relatively slower myosin attachment kinetics in the bag fiber and a more compliant passive elastic element (*Gladden and Boyd, 1985*; *Gladden, 1976*; *Thornell et al., 2015*). Proske (*Proske et al., 1992*; *Proske et al., 2014*; *Proske and Gandevia, 2012*) hypothesized that history-dependent muscle spindle IFRs (and corresponding perceptual errors) are caused by a population of cross-bridges within the intrafusal muscle that are unable to 'keep up' with the rate of shortening during an imposed movement, causing the intrafusal muscle fibers to fall slack. To model this behavior, we selected rate equations that would produce relatively slow cross-bridge reattachment during shortening, but would retain other desired characteristics, such as short-range stiffness.

## Model of muscle spindle responses to stretch of intrafusal muscle

To model the transformation of intrafusal muscle fiber stress into a firing waveform, we used a pseudolinear combination of force and its first time-derivative, yank, based on previously published observations (*Blum et al., 2017*). Our model consists of two intrafusal muscle fiber models, a 'static' fiber and a 'dynamic' fiber based on observations that muscle spindle primary afferent responses to stretch consist of two components (*Banks et al., 1997*; *Blum et al., 2017*; *Boyd, 1976*; *Boyd et al., 1977*; *Hasan, 1983*; *Jami et al., 1982*; *Jami and Petit, 1979*; *Lewis and Proske, 1972*). For these simulations, each muscle fiber model used identical parameters, but the contribution of each fiber to the neural firing rate varied. The equation describing the contribution of each fiber to the total firing rate is:

$$r(t) = r_{dynamic}(t) + r_{static}(t),$$

where the total firing rate of the afferent, $r(t)$, is a sum of the dynamic and static fiber components, or $r_{dynamic}(t)$ and $r_{static}(t)$, respectively. The static component was defined as:

$$r_{static}(t) = k_{Fs} \, F_s(t),$$

where $k_{Fs}$ is a constant and $F_s(t)$ is the total force in the static fiber. The dynamic component was defined as:

$$r_{dynamic}(t) = k_{Fd} \, F_d(t) + k_{\dot{F}d} \, \dot{F}_d(t),$$

where $k_{Fd}$ and $k_{\dot{F}d}$ are constants, and $F_d(t)$ and $\dot{F}_d(t)$ are respectively the force and yank of the cycling cross-bridges in the dynamic fiber. We used default $k_{Fs}$, $k_{Fd}$, and $k_{\dot{F}d}$ values of 1, 1, and 0.03, respectively, unless otherwise noted.

The static and dynamic fibers are arranged in perfect mechanical parallel and were allowed to be activated independently. Thus, the actions of the dynamic and static fibers could be simulated simultaneously or sequentially.

## Occlusion between dynamic and static components

To account for the evidence of so-called 'occlusive interaction' between dynamic and static branches of the muscle spindle Ia afferent ending, we used a nonlinear summation of the static and dynamic components. Previous models have used complete occlusion (*Hulliger et al., 1977*; *Lin and Crago, 2002*) but we used a partial occlusion based on more recent findings (*Banks et al., 1997*). With occlusion, the total firing rate of the model Ia afferent becomes:

$$r(t) = f_{occ}r_{dynamic}(t) + r_{static}(t), \quad r_{dynamic} \geq r_{static}$$

$$r(t) = r_{dynamic}(t) + f_{occ}r_{static}(t), \quad r_{static} > r_{dynamic}$$

where $f_{occ}$ is an occlusion factor limiting the contribution of either component to the overall firing

rate. This parameter was set to 0.3 (unitless) for all simulations unless otherwise noted (*Banks et al., 1997*).

## Dynamic response simulations

To demonstrate the ability of our model to produce the classical fractional power relationship between the dynamic response of muscle spindle Ia firing rates and ramp velocity (*Hasan, 1983*; *Houk et al., 1981*; *Matthews, 1963*), we applied a series of ramp-hold stretches to the model with each fiber's proportion of available binding sites set to 0.3 (*Figure 6A*). The ramp stretches consisted of a pre-stretch isometric hold period, followed by a constant velocity stretch that varied linearly between trials from $0.079L_0$/s to $0.79L_0$/s, and another isometric hold period at its new length ($1.059L_0$). The duration of stretch was shortened proportionally to the stretch velocity to ensure the same total length was applied in each trial.

## Time-history dependence simulations

To demonstrate the unique ability of our model to vary its own sensitivity to stretch based on the history of movement applied to the muscle (*Haftel et al., 2004*), we applied series of triangular ramp-release stretches with each fiber's activation set to 0.3. Each series consisted of three stretch-shorten cycles, with a $1.047L_0$ amplitude and stretch and shorten velocities of $0.079L_0$/s. The first two cycles were applied sequentially with no pause between them, whereas the third sequence was applied after a varied isometric hold period at $L_0$ ranging from 0 to 10 s.

To demonstrate the robustness of our mechanistic model to produce history dependent responses similar to those observed in awake humans, we approximated stimuli from microneurography studies in humans from the lower limb (*Day et al., 2017*). We applied a $1.042 L_0$ sinusoidal length change at 1.57 Hz to the model at the baseline activation to mimic the passive manipulation of the ankle in the study (*Day et al., 2017*). To roughly match the predicted driving potential to the firing rate of the spindle, we used $k_{Fs}$, $k_{Fd}$, and $k_{\dot{F}d}$ values of 1.8, 2, and 0.15, respectively.

## Tonic gamma activation simulations

To demonstrate the effects of muscle activation on the firing response of our model (*Figure 7*), we applied a range of activations to the static and dynamic fibers (*Emonet-Dénand et al., 1977*). We varied the activation levels of the static and dynamic fibers independently, between 0–1.0, before applying a $1.047L_0$ ramp-hold stretch at a constant velocity of $0.079L_0$/s. We used $k_{Fs}$, $k_{Fd}$, and $k_{\dot{F}d}$ values of 1.5, 0.8, and 0.03, respectively, for these simulations in order to better visualize the effects of gamma activation on the predicted driving potential.

## Musculotendon stimulations

To more completely address the complex interplay between alpha and fusimotor drive and the effect therein on the afferent outputs of the spindle, we created a model of a musculotendinous unit, comprised of a single half-sarcomere model mechanically in series with a linear elastic element to model the tendon. To balance the forces between the muscle and tendon, a force balancing operation was performed by searching iteratively for the pair of muscle and tendon length changes that would result in equal tendon and muscle forces.

To demonstrate the independent effects of alpha and fusimotor drive on the muscle spindle Ia afferent, we modeled a series of isometric activation pulses. Each pulse was based on a Gaussian waveform (*gausswin.m*) 1 s long, with a width factor of 0.3. Medium activation was defined as 50% peak activation, and low activation was defined as 25% peak activation for both extra- and intrafusal muscle. We simulated the musculotendon responses to these pulses and used the extrafusal muscle length as the length input to the intrafusal muscle fibers. We calculated the resulting firing rate using the previously defined *r*, from the force and yank of intrafusal models. As a visual aide, we generated a possible Ia afferent spike response by driving a leaky integrate-and-fire neural model with *r* as the input.

Next, we wanted to demonstrate the opposing effects of extrafusal muscle shortening and increasing fusimotor drive to mimic the effects of putative alpha-gamma coactivation during isometric conditions. To accomplish this, we emulated the microneurography work of *Edin and Vallbo, 1990* by applying ramp increases to extrafusal activation with concomitant fusimotor ramps. We ran

16 simulations with increasing 2 s activation ramps with peak activation at the end of the ramp ranging from 25–100% extrafusal activation in linear increments. The intrafusal peak activation was 100% for all 16 simulations.

Finally, to demonstrate the combined interactions of alpha and fusimotor drive with musculotendon movements, we ran a set of simulations varying levels of extra- and intrafusal activation during an imposed stretch of the musculotendon. For each simulation, we applied a 3.8% $L_0$, 7.7% $L_0$/s ramp-hold stretch to the musculotendon under low (10%) and medium (60%) tonic extrafusal activation. For each of these conditions, we also simulated low (30%) and high (70%) tonic dynamic and static intrafusal activation to highlight the effects of extrafusal activation on Ia afferent coding. As with the isometric pulse simulations, we visualized Ia afferent spiking using a leaky integrate-and-fire neuron.

### Adaptation of previously published figures

Previously published results that were used for comparison with our model predictions were redrawn in Adobe Illustrator. Only single data points and lines were approximated by tracing over their apparent geometric centroids. These data were redrawn for aesthetic purposes only and were not used for any quantitative comparisons. Any comparison of data from these studies with the present study were performed using the original manuscripts.

## Additional information

### Funding

| Funder | Grant reference number | Author |
|--------|----------------------|--------|
| Eunice Kennedy Shriver National Institute of Child Health and Human Development | R01 HD90642 | Kenneth S Campbell Timothy C Cope Lena H Ting |
| National Cancer Institute | R01 CA221363 | Timothy C Cope |
| National Institute of Neurological Disorders and Stroke | F31 NS093855 | Kyle P Blum |
| Government of Canada | BPF-156622 | Brian C Horslen |

The funders had no role in study design, data collection and interpretation, or the decision to submit the work for publication.

### Author contributions

Kyle P Blum, Conceptualization, Data curation, Software, Formal analysis, Funding acquisition, Validation, Investigation, Visualization, Methodology, Writing - original draft, Writing - review and editing; Kenneth S Campbell, Conceptualization, Resources, Software, Formal analysis, Supervision, Funding acquisition, Validation, Investigation, Visualization, Methodology, Writing - original draft, Project administration, Writing - review and editing; Brian C Horslen, Timothy C Cope, Software, Formal analysis, Validation, Investigation, Visualization, Methodology, Writing - original draft, Writing - review and editing; Paul Nardelli, Conceptualization, Data curation, Funding acquisition, Investigation, Methodology, Writing - original draft, Project administration, Writing - review and editing; Stephen N Housley, Conceptualization, Data curation, Formal analysis, Investigation, Methodology, Writing - review and editing; Lena H Ting, Conceptualization, Resources, Data curation, Supervision, Funding acquisition, Investigation, Visualization, Methodology, Writing - original draft, Project administration, Writing - review and editing

### Author ORCIDs

Kyle P Blum (iD) https://orcid.org/0000-0002-9760-2053
Lena H Ting (iD) https://orcid.org/0000-0001-6854-9444

### Ethics

Animal experimentation: All procedures and experiments were approved by the Georgia Institute of Technology's Institutional Animal Care and Use Committee. Adult female Wistar rats (250-300 g) were studied in terminal experiments only and were not subject to any other experimental procedures. All animals were housed in clean cages and provided food and water ad libitum in a temperature- and light-controlled environment in Georgia Institute of Technology's Animal facility (protocol A16038).

### Decision letter and Author response

Decision letter https://doi.org/10.7554/eLife.55177.sa1
Author response https://doi.org/10.7554/eLife.55177.sa2

## Additional files

### Supplementary files

• Supplementary file 1. Constant parameters used in both dynamic and static intrafusal muscle fiber models. These parameters did not change in any simulation presented in this study. When two values are presented, they represent the respective values for the dynamic and static fibers.

### Data availability

Code has been made available on GitHub: https://github.com/kylepblum/MechanisticSpindleManuscript (copy archived at https://archive.softwareheritage.org/swh:1:rev:da0ed89078a948167-b4e2b511480787ddb681892/). Data have been made available on Dryad: https://doi.org/10.5061/dryad.vdncjsxsw.

The following dataset was generated:

| Author(s) | Year | Dataset title | Dataset URL | Database and Identifier |
|---|---|---|---|---|
| Blum KP, Campbell KS, Horslen BC, Nardelli P, Housley SN, Cope TC, Ting LH | 2020 | Data from: Diverse and complex muscle spindle afferent firing properties emerge from multiscale muscle mechanics | https://doi.org/10.5061/dryad.vdncjsxsw | Dryad Digital Repository, 10.5061/dryad.vdncjsxsw |

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
