## [Decision Letter]

**Acceptance summary:**

This mammalian muscle spindle is a central player in the sense of proprioception. It is a complex biological device that is generally sensitive to changes in the length of the skeletal muscle within which it resides. Yet it's response properties (i.e., firing rates of its sensory afferents) are oftentimes inexplicable in terms of it operating as a simple muscle-length sensor. This paper represents a major step forward in understanding the basis of the elegant intricacies of the muscle spindle. The authors have used a cross-bridge based model of the intrafusal fibers of the muscle spindle to accurately predict (and provide insight into) Ia afferent responses to a host of mechanical stimuli. As illuminated by the simulations, Ia afferent activity is shaped by interactions among multiple mechanical and neural factors. While some questions remain as to the nature of the transduction process itself, this work offers a key advance in deciphering the enigmatic muscle spindle.

**Decision letter after peer review:**

Thank you for submitting your article "Diverse muscle spindle firing properties emerge from multiscale muscle mechanics" for consideration by *eLife*. Your article has been reviewed by three peer reviewers, including Andrew Fuglevand as the Reviewing Editor and Reviewer #1, and the evaluation has been overseen by John Huguenard as the Senior Editor. The following individuals involved in review of your submission have agreed to reveal their identity: Arthur Prochazka (Reviewer #2); Gerald Loeb (Reviewer #3).

The reviewers have discussed the reviews with one another and the Reviewing Editor has drafted this decision to help you prepare a revised submission.

As the editors have judged that your manuscript is of interest, but as described below that additional simulations are required before it is published, we would like to draw your attention to changes in our revision policy that we have made in response to COVID-19 (https://elifesciences.org/articles/57162). First, because many researchers have temporarily lost access to the labs, we will give authors as much time as they need to submit revised manuscripts. We are also offering, if you choose, to post the manuscript to bioRxiv (if it is not already there) along with this decision letter and a formal designation that the manuscript is 'in revision at *eLife*'. Please let us know if you would like to pursue this option. (If your work is more suitable for medRxiv, you will need to post the preprint yourself, as the mechanisms for us to do so are still in development.)

Summary:

The muscle spindle is one of the most thoroughly studied sensory receptors in the somatosensory system, yet much is still unknown about how it works. Commendably, the authors have attempted to model the responses of spindle sensory afferents using a biophysical model of intrafusal muscle fibers. The model was shown to mimic experimentally recorded afferent activity in a number of situations. Indeed, it is encouraging to see attention being paid again to the elegant complexities of spindle receptors after years of over-simplification in control models. Nevertheless, there are concerns (detailed in the essential revisions below) about those aspects that were left out.

Essential revisions:

1) The assumption that extrafusal muscle force can serve as a proxy for intrafusal fiber force needs to be fully addressed. Indeed, there are well known situations for which an assumed correspondence between extrafusal and intrafusal forces would seem to fail to reproduce experimental results. For example, the classical experimental signature used to identify Ia afferents is a cessation in their discharge during an evoked twitch in the extrafusal muscle fibers. Likewise, the model would seem to fail to reproduce spindle afferent responses during imposed length changes with and without concomitant homonymous extrafusal muscle contractions (e.g. Elek, Prochazka, Hulliger, Vincent. In-series compliance of gastrocnemius muscle in cat step cycle: do spindles signal origin-to-insertion length? J. Physiol., 429, 237-258, 1990). The authors need to include additional simulations of these fundamental experimental phenomena and to fully address the outcome in the Discussion.

2) The authors suggest that their model provides a unifying biophysical framework for understanding muscle spindle activity, yet there was little attention paid to how intrafusal force or yank is transduced into a receptor potential. Such a unifying framework would need to include mechanisms of transduction by mechanically-gated ion channels. As such, the role that sensory transduction mechanisms play in shaping spindle afferent activity needs to be addressed – either in the model or in the Discussion.

3) The role that intrinsic properties and associated time-varying conductances (e.g. such as those underlying spike-frequency adaptation) in muscle spindle afferents may play in influencing firing dynamics needs to be addressed in the model or in the Discussion.

4) There needs to be more clarity in the description of the model and what aspects of the model were original and what aspects were based on previous work, for example, that of Campbell et al. (2014) and MyoSim.

5) The simulated response (i.e. the driving potential) of the biophysical model depicted in Figure 6A to repeated triangular length changes (without pauses) does not resemble the experimental firing rate data to repeated triangular length changes shown in Figure 2B. In particular, the model exhibits marked abbreviation of the responses to the 2nd and 3rd length changes that are not evident in the experimental data of Figure 2. This disparity between experimental and simulated findings needs to be discussed.

6) Any general model that aims to account for the activity of spindle afferents during natural activities must account for the well-documented independence among α, γ dynamic and γ static activation patterns and kinematics, whose different effects on Ia activity have been simulated, measured or inferred in a variety of experiments and integrated into previous models (see Mileusnic et al., 2006). The Discussion should identify which scenarios have not been simulated and which might be problematic for their general thesis.

---

## [Author Response]

Essential revisions:1) The assumption that extrafusal muscle force can serve as a proxy for intrafusal fiber force needs to be fully addressed. Indeed, there are well known situations for which an assumed correspondence between extrafusal and intrafusal forces would seem to fail to reproduce experimental results. For example, the classical experimental signature used to identify Ia afferents is a cessation in their discharge during an evoked twitch in the extrafusal muscle fibers.

We thank the reviewers for these remarks and wholeheartedly agree. We acknowledge that the similarities between intrafusal and extrafusal muscle fibers are limited to the passive conditions. These similarities have been profoundly useful in reinterpreting many aspects of muscle spindle data obtained in such conditions, leading to the insight that the complex properties of the intrafusal muscle itself need to be considered in muscle spindle firing. We have now made it clearer throughout the text that the similarity of extrafusal and intrafusal force is limited to passive stretch conditions. Here are a few examples:

“In the relaxed condition, i.e. in the absence of central drive to the muscles, we assumed that extrafusal muscle fiber forces provide a reasonable proxy for resistive forces of the intrafusal muscle fibers within the muscle spindle mechanosensory apparatus (Figure 1).”

“We assumed average extrafusal fiber force to be proportional to intrafusal muscle fiber force in the anesthetized, passive stretch condition only.”

“During the swing phase of the cat locomotor step cycle, relaxed ankle extensor muscles are stretched by activity in ankle flexor muscles, and extensor Ia firing rates appear to follow muscle velocity and/or length (A Prochazka and Gorassini, 1998; Arthur Prochazka and Gorassini, 1998), which closely resemble extrafusal and intrafusal muscle force and yank (Blum et al., 2017) in passive conditions.”

Likewise, the model would seem to fail to reproduce spindle afferent responses during imposed length changes with and without concomitant homonymous extrafusal muscle contractions (e.g. Elek, Prochazka, Hulliger, Vincent. In-series compliance of gastrocnemius muscle in cat step cycle: do spindles signal origin-to-insertion length? J. Physiol., 429, 237-258, 1990). The authors need to include additional simulations of these fundamental experimental phenomena and to fully address the outcome in the Discussion.

To address the reviewer comments about conditions in which extrafusal muscle contractions occur, we have developed new simulations where we embedded the biophysical muscle spindle model within a musculotendon unit, providing a more generalizable platform with which to explore the effects of independent extrafusal and intrafusal activation. We have included simulations that explicitly simulate conditions in which the intra- and extrafusal force differ, and demonstrate how muscle spindle firing following intrafusal muscle fiber force could explain prior experimental conditions. We include cases where extrafusal muscle contractions occur during both isometric (Figure 8) and muscle stretch conditions (Figure 9). Indeed, as shown in the Elek et al. paper, the muscle spindle Ia firing ceases when extrafusal muscles are contracted without a concomitant change in γ drive. These simulations enabled us to clearly illustrate that neither extrafusal force nor length can independently–or together–account for muscle spindle firing patterns, Instead, our model illustrates that the intrafusal force and yank arise from a complex interaction between muscle, tendon, load, and intrafusal muscle mechanics.

The end of the Results have been thoroughly revised to discuss newly generated Figures 8 and 9.

2) The authors suggest that their model provides a unifying biophysical framework for understanding muscle spindle activity, yet there was little attention paid to how intrafusal force or yank is transduced into a receptor potential. Such a unifying framework would need to include mechanisms of transduction by mechanically-gated ion channels. As such, the role that sensory transduction mechanisms play in shaping spindle afferent activity needs to be addressed – either in the model or in the Discussion.

We appreciate this comment. We fully acknowledge that we have focused on the relationship between the mechanical properties and signals within the muscle in relation to muscle spindle receptor potentials and firing patterns without providing a biophysical substrate for the mechnotransduction process. It is our goal to eventually incorporate these critical components into the model, and have begun to do so in collaboration with Dr. Mark Binder. Unfortunately, we currently lack sufficient information about the properties of the encoding region, the ion channels within the afferent fiber, and their interactions to develop such a model. Dr. Cope’s current work is indeed focused on identifying ion channels in the muscle spindle afferents, as is the work of several other researchers such as Guy Biewick. We still lack properties of the equatorial regions and recordings of muscle spindle kinetics and kinematics during muscle spindle firing.

We have now included an explicit section on limitations of the model, and included references to highlight the paucity of adequate data regarding the potential mechanisms transforming forces within the intrafusal fiber into a receptor potential in the muscle spindle afferent throughout the text, particularly:

“A comprehensive understanding of the mechanisms underlying muscle spindle mechanotransduction, i.e. translation of mechanical stimuli to receptor potentials, remains elusive. Lacking sufficiently detailed information from the literature, we represented the entire neuromechanical transduction process by a set of constant gains. However, we do recognize that the dynamics of yet uncharacterized viscoelastic properties of the equatorial regions of intrafusal muscle fibers, ion channels in muscle spindle afferent endings (Bewick and Banks, 2015; Carrasco et al., 2017), and intrinsic neural dynamics (e.g. those underlying neural history dependence, spike-frequency adaptation, and occlusion Banks et al., 1997) of the afferent plays a significant role in the resulting muscle spindle firing signals.”

We have also removed reference to the “unifying” framework, but we still argue that this initial biophysical framework is an important foundation for developing computational models of the mechanotrasnductive process. Having established that the firing properties of the spindle Ia afferent are in large part related to intrafusal muscle fiber force and yank should provide constraints on the inputs to and outputs of the poorly understood mechanotransduction apparatus in the muscle spindle.

“The core framework presented provides proof of concept for more elaborate and biophysically accurate model of muscle spindle firing necessary to predict muscle spindle signals during naturalistic movement conditions in health and disease. Importantly, our model can be used as a platform to enable the development of computational models capable of testing the effects of a wide variety of multiscale mechanisms on muscle spindle firing, including architectural arrangement of the muscle spindle within the muscle (Maas et al., 2009), intra- and extrafusal muscle myosin expression, more complex muscle force-generating mechanisms, extracellular matrix stiffness, mechanosensory encoding mechanisms, and biophysical neural dynamics that could all be affected by aging and disease.”

3) The role that intrinsic properties and associated time-varying conductances (e.g. such as those underlying spike-frequency adaptation) in muscle spindle afferents may play in influencing firing dynamics needs to be addressed in the model or in the Discussion.

We agree that the neural dynamics could significantly shape the firing of the muscle spindle, which we now address in the Discussion. Indeed, we have been surprised by the degree to which our estimates of intrafusal force based on whole muscle force in passive conditions, but these will likely change in more complex conditions. We have modified the text to more fully acknowledge the limitation of our model and other sources of variability.

4) There needs to be more clarity in the description of the model and what aspects of the model were original and what aspects were based on previous work, for example, that of Campbell et al. (2014) and MyoSim.

We have added substantially more detail about the construction of the intrafusal muscle models. We also clarified within this text how our model differs from MyoSim, but we will further clarify here:

MyoSim was developed by Campbell as a software package for researchers to model muscle based on the dynamic coupling of thick and thin filaments within sarcomeres. Because we wanted to test a hypothesis primarily regarding myosin kinetics, we implemented only this portion of simulation in MATLAB, and simplified the thin filament activation to be an input chosen by the user.

The kinetic scheme was designed specifically for this study to test whether the mechanical properties of intrafusal muscle fiber could account for the wide range of muscle spindle Ia afferent firing rates observed in the experiments we outlined in the manuscript in the extensively updated text of the Materials and methods.

5) The simulated response (i.e. the driving potential) of the biophysical model depicted in Figure 6A to repeated triangular length changes (without pauses) does not resemble the experimental firing rate data to repeated triangular length changes shown in Figure 2B. In particular, the model exhibits marked abbreviation of the responses to the 2nd and 3rd length changes that are not evident in the experimental data of Figure 2. This disparity between experimental and simulated findings needs to be discussed.

We have added a limitations section and acknowledgement of the disparities and the need to increase the fidelity of the model.

“It should be noted that, despite the discrepancies in the more abrupt stretch-shorten “triangle” stretches (Figure 6A), our model was able to closely reproduce the human Ia afferent response (Figure 6B, C) during the slow sinusoidal stretch, which may be more physiologically relevant. […] Another possibility is that we failed to account for the complex process(es) responsible for the occlusion between multiple transduction zones in the primary afferent (Banks et al., 1997; Mileusnic et al., 2006), in which the firing rate may be more dependent on the bag1 fiber during the initial stretch and become more dependent on to bag2 and chain fibers in later stretches.”

6) Any general model that aims to account for the activity of spindle afferents during natural activities must account for the well-documented independence among α, γ dynamic and γ static activation patterns and kinematics, whose different effects on Ia activity have been simulated, measured or inferred in a variety of experiments and integrated into previous models (see Mileusnic et al., 2006). The Discussion should identify which scenarios have not been simulated and which might be problematic for their general thesis.

We believe that our new simulation better address the independence of α and γ drive in movement. We have now included language in the Discussion to emphasize that our simulations are not comprehensive, particularly in the context of:

Scenarios simulated in this paper:

“We demonstrated that a diverse range of Ia afferent firing characteristics during passive stretch – well-documented in the literature but not completely understood mechanistically – is emergent from first principles of intrafusal muscle contractile (cross-bridge) mechanisms. […] Finally, we simulated multiscale mechanical interactions including movements, α activation of extrafusal muscle, and γ activation of intrafusal muscle to demonstrate the importance of these complex interactions when modeling proprioceptive activity.”

Some scenarios we did not simulate which may be problematic:

“The multiscale model presented here provides a mechanistic framework that can explain the rich diversity of movement-related biomechanical signals in naturalistic behaviors. […] Simulating such intermuscular interactions is likely necessary to predict proprioceptor activity during movement, and would require multiple muscle and muscle spindle models to be coupled.”